# Assessment of Water Absorption Capacity and Cooking Time of Wild Under-Exploited *Vigna* Species towards their Domestication

**Difo Voukang Harouna** [1,2], **Pavithravani B. Venkataramana** [2,3], **Athanasia O. Matemu** [1] and **Patrick Alois Ndakidemi** [2,3,*]

[1] Department of Food Biotechnology and Nutritional Sciences, Nelson Mandela African Institution of Science and Technology (NM-AIST), Arusha P.O. Box 447, Tanzania

[2] Centre for Research, Agricultural Advancement, Teaching Excellence and Sustainability in Food and Nutrition Security (CREATES-FNS), Nelson Mandela African Institution of Science and Technology, Arusha P.O. Box 447, Tanzania

[3] Department of Sustainable Agriculture, Biodiversity and Ecosystems Management, Nelson Mandela African Institution of Science and Technology, Arusha P.O. Box 447, Tanzania

[*] Correspondence: Patrick.ndakidemi@nm-aist.ac.tz; Tel.: +255-757-744-772

**Abstract:** Some phenotypic traits from wild legumes are relatively less examined and exploited towards their domestication and improvement. Cooking time for instance, is one of the most central factors that direct a consumer's choice for a food legume. However, such characters, together with seed water absorption capacity are less examined by scientists, especially in wild legumes. Therefore, this study explores the cooking time and the water absorption capacity upon soaking on 84 accessions of wild *Vigna* legumes and establishes a relationship between their cooking time and water absorbed during soaking for the very first time. The accessions were grown in two agro-ecological zones and used in this study. The Mattson cooker apparatus was used to determine the cooking time of each accession and 24 h soaking was performed to evaluate water absorbed by each accession. The two-way analysis of variance revealed that there is no interaction between the water absorption capacity and cooking time of the wild *Vigna* accessions with their locations or growing environments. The study revealed that there is no environment × genotype interaction with respect to cooking time and water absorption capacity as phenotypic traits while genotype interactions were noted for both traits within location studied. Furthermore, 11 wild genotypes of *Vigna* accessions showed no interaction between the cooking time and the water absorption capacity when tested. However, a strong negative correlation was observed in some of the wild *Vigna* species which present phenotypic similarities and clusters with domesticated varieties. The study could also help to speculate on some candidates for domestication among the wild *Vigna* species. Such key preliminary information could be of vital consideration in breeding, improvement, and domestication of wild *Vigna* legumes to make them useful for human benefit as far as cooking time is concerned.

**Keywords:** non-domesticated legumes; *Vigna racemosa*; *Vigna ambacensis*; *Vigna reticulata*; *Vigna vexillata*; wild food legumes; legumes; *Vigna* species; cooking time; water absorption; domestication

## 1. Introduction

Legumes (family: Fabaceae), the third largest family among flowering plants, grouping about 650 genera and 20,000 species, represent the second most valuable plant source of nutrients for both humans and animals [1]. Their importance in human life through positive impact in global food security is uncontestable due to the contribution of some of the domesticated commercialized legumes

such as soybeans, cowpeas, and common beans [2]. Yet, their production rate remains unsatisfying compared with their consumption rate due to several challenges ranging from agronomic constraints to policy issues through farmers' and consumers' acceptability [3,4]. These challenges have directed the interest of some scientists towards investigating novel alternatives by screening the hitherto wild non-domesticated species within the little-known genera of legumes in order to find important traits that fit consumers' acceptance and desire without necessarily genetically engineering them [4].

Generally, taste and smell are the first senses that come to peoples' mind whenever they think about the consumption of food and drink [5]. However, consumers' responses to food depend on several factors not only limited to sensory characteristics of the product and their physiological status. They also depend on other factors, such as previous information acquired about the product, their past experience, and their attitudes and beliefs [6].

Soaking is usually a processing technique performed prior to cooking of grains and legumes. Hence, the evaluation of water absorption characteristics of different seeds during soaking is an important parameter that is well considered by researchers who have proven that grains show different water absorption rates and water absorption capacities in different soaking conditions [7]. Understanding water absorption in legumes during soaking is a very important aspect because it affects succeeding processes such as the cooking time and the quality of the final product [8]. Water absorption of seeds during soaking mainly depends on soaking time, water temperature, and some seeds' physical characteristics like hardness and seed coat thickness, and may be related to cooking time for a specific type of grain or legume. This is one of the gaps that this study is attempting to address.

Cooking time, a sign of cooking quality, is one of the most central factors that direct a consumer's choice for a food legume as longer cooking time is one of the foremost limitations that make legumes uneconomical and unacceptable to consumers [9]. Cooking usually implies heat application that causes physicochemical changes like gelatinization of starch, denaturation of proteins, solubilization of some of the polysaccharides, softening and breakdown of the middle lamella, a cementing material found in the cotyledon [9]. Cooking also inactivates or reduces the levels of anti-nutrients such as trypsin inhibitors and flatulence-causing oligosaccharides, resulting in improved nutritional and sensory qualities [10].

Cooking time is also one of the phenotypic characters assessed by many breeding programs using the Mattson Bean Cooker as the recommended equipment for measuring the variable [11]. The cooking time of legumes depend on their genera, species, and varieties [7,9].

Common beans (*Phaseolus vulgaris*) and cowpea (*Vigna unguiculata*) are the mainly cultivated and consumed varieties of legumes worldwide that belong to two different genera, the *Phaseolus* and *Vigna* respectively [2,12,13]. It is reported that fewer domesticated edible species as compared with the numerous non-domesticated wild species exist in most legumes' genera. Domesticated and semi-domesticated species are denoted as neglected and underutilized species due to little attention being paid to them or the complete ignorance of their existence by agricultural researchers, plant breeders, and policymakers [14].

The genus *Vigna*, of the present study, is a large collection of vital legumes consisting of more than 200 species [15]. It comprises several species of agronomic, economic, and environmental importance. The most common domesticated ones include the mung bean (*V. radiata* (L.) Wilczek), urd bean (*V. mungo* (L.) Hepper), cowpea (*V. unguiculata* (L.) Walp.), azuki bean (*V. angularis* (Willd.) Ohwi & Ohashi), bambara groundnut (*V. subterranea* (L.) Verdc.), moth bean (*V. aconitifolia* (Jacq.) Maréchal), and rice bean (*V. umbellata* (Thunb.) Ohwi & Ohashi). Many of these species are valued as forage, green manure, and cover crops, besides their value as high protein grains. Moreover, the genus also comprises more than 100 wild species that do not possess common names apart from their scientific appellation yet [16]. They are simply known as underexploited wild *Vigna* species, or non-domesticated *Vigna* species [2,15]. This could be some of the reasons as to why very little or almost no scientific attention has been given to them, especially concerning the human domestic utilization such as consumption, cookability, functional, and processing characteristics such as water absorption capacity and soaking.

Therefore, this study evaluates the cooking time and water absorption capacity upon soaking on 84 accessions of wild *Vigna* legumes and establishes a relationship between their cooking time and water absorbed during soaking.

## 2. Materials and Methods

### 2.1. Sample Collection and Preparation

The 84 accessions of wild *Vigna* species (seeds) used in this study were obtained from gene banks as presented in Table 1. All the accessions were planted in an experimental plot following the augmented block design arrangement [17] and allowed to grow until complete maturity before harvesting in order to have enough seeds for analysis. After harvesting, the seeds were sun-dried to maintain a uniform moisture content of grains to 10%–14% using a moisture grain tester (DICKEY-JOHN, model: MINIGAC 1, Minneapolis, MN, USA) [18]. An illustration of the seeds of some of the samples is also shown in Figure 1. In addition, three domesticated *Vigna* legumes—that is, cowpea (*V. unguiculata*), rice bean (*V. umbellata*), and a semi-domesticated landrace (*V. vexillata*)—were used as checks. The checks were obtained from the Genetic Resource Center (GRC-IITA), Nigeria (cowpea), the National Bureau of Plant Genetic Resources (NBPGR), India (rice bean), and the Australian Grain Genebank (AGG), Australia (semi-domesticated landrace *V. vexillata*).

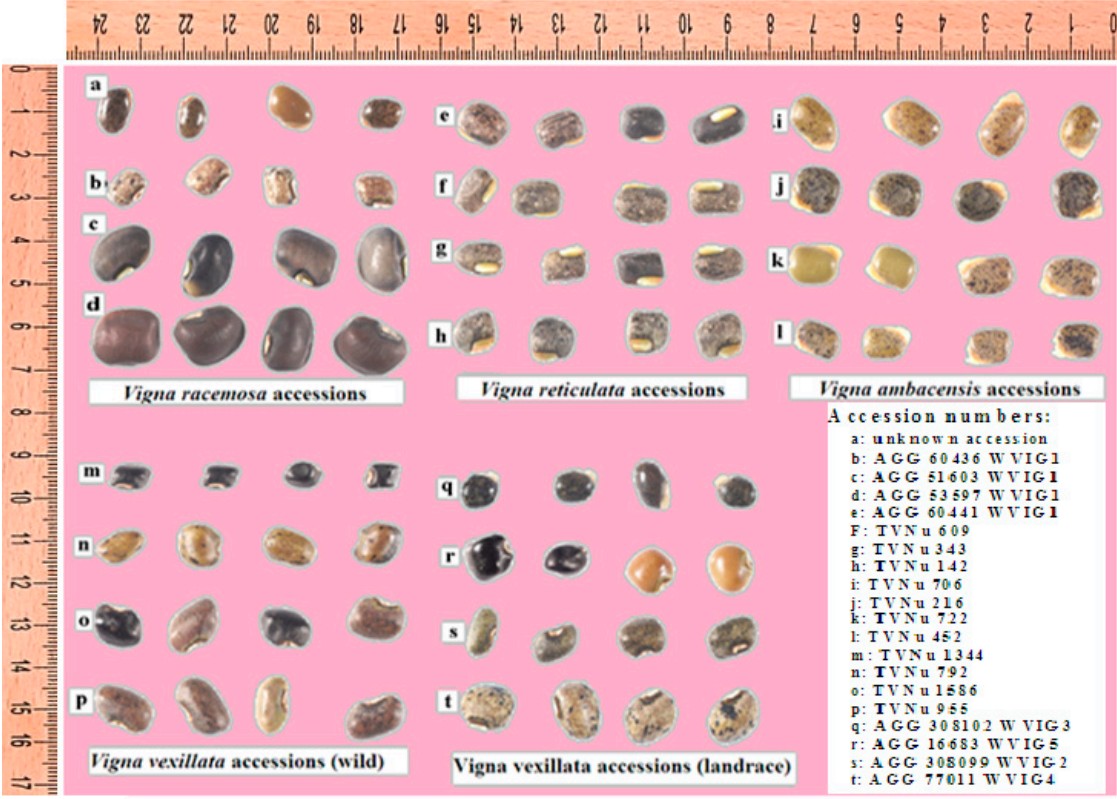

**Figure 1.** Photographs illustrating seed morphology of wild *Vigna* species. Four seeds per accession were pictured under the same conditions to give an image of the morphology and the relative size. Distances of lines in the background are 1 cm in the vertical and horizontal directions. Source: Authors based on seeds requested from the Australian Grain Genebank (AGG) (**a–e,q–t**) and the Genetic Resources Center, International Institute of Tropical Agriculture, (IITA), Ibadan, Nigeria (**f–p**).

**Table 1.** Wild *Vigna* species collected from the gene banks/self.

| *Vigna* Species | Genebank/Number of Accession | | | Total |
|---|---|---|---|---|
| | GRC, IITA Ibadan, Nigeria | AGG Horsham, Victoria | Self-Collected | |
| *Vigna racemosa* | - | 4 | - | 4 |
| *Vigna reticulata* | 30 | 1 | - | 31 |
| *Vigna vexillata* | 29 | 6 | - | 35 |
| *Vigna ambacensis* | 11 | 0 | - | 11 |
| *Unknown V. racemosa Accession (Nigeria)* | - | - | 1 | 1 |
| *Unknown V. reticulata Accession (Nigeria)* | - | - | 1 | 1 |
| *Unknown Vigna (Tanzania)* | - | - | 1 | 1 |
| Total | 70 | 11 | 3 | 84 |

GRC, IITA: Genetic Resource Center, Germplasm Health Unit, International Institute of Tropical Agriculture (IITA), Headquarters, PMB 5320, Oyo Road, Idi-Oshe, Ibadan, Nigeria. AGG: Australian Grain Genebank, Department of Economic Development, Jobs, Transport and Resources, Private Bag 260, Horsham, Victoria 3401.

## 2.2. Sample Cultivation (Multiplication) Process: Experimental Design and Study Site.

The collected seeds were planted in two agro-ecological zones located at two agricultural research stations in Tanzania during the main cropping season (March–September, 2018). The first site was at the Tanzania Coffee Research Institute (TaCRI), located at Hai District, Moshi, Kilimanjaro region (latitude 3°13′59.59″ S, longitude 37°14′54″ E) which is at a high altitude (1681 m) a.s.l. The second site was at the Tanzania Agricultural Research Institute (TARI), Selian, Arusha, which is at a mid-altitude agroecological zone. TARI-Selian lies at latitude 3°21′50.08″ N and longitude 36°38′06.29″ E at an elevation of 1390 m a.s.l.

A total of 160 accessions of wild *Vigna* legumes were planted in an augmented block design following the randomization generated by the statistical tool on the website [19] with three checks. The seeds were planted in eight blocks of 26 lines each with every line containing 10 seeds of each accession. Each check was replicated two times in a block as generated by the statistical tool. The field was monitored and maintained in good conditions from germination to complete maturity before harvesting and seeds were prepared for further analysis. Eighty-four accessions were then selected based on the availability of seeds after maturity for this study.

## 2.3. Seed Soaking Process

The soaking method adopted from McWatters and modified by Shafaei was used with a slight modification [7,20].

Ten seeds of each accession were randomly selected and weighed, then placed in glass beakers containing 200 mL distilled water and allowed to stand at room temperature (25 °C) for 24 h. The weight of water absorbed by various seeds was measured after 24 h, as it is the soaking time generally practiced by most consumers at home. After reaching required time, the soaked samples were removed from the beakers and placed on a blotter paper to eliminate the excess water, and then weighed. A precision electronic balance (Model GF400, accuracy ± 0.001 g A&D Company Ltd, Taunton, MA, USA) was used to measure weight of sample before and after immersion. All tests were performed in triplicate. The weight of water absorbed was determined using the formula below [7,20]:

$$Wa = (Wf - Wi)/Wi,$$

where, *Wa* is the water absorption, *Wf* is weight of seeds after immersion (g), and *Wi* is weight of seeds before immersion (g).

### 2.4. Cooking Process on a Mattson Bean Cooker

A Mattson Bean Cooker (MBC) apparatus was used to record the mean cooking time of each accession of wild *Vigna* legume. The apparatus consists of 25 plungers and a cooking rack with 25 reservoir-like perforated saddles, each of which holds a grain and a plunger calibrated to a specific weight. Each plunger weighs 90 g and terminates in a stainless-steel probe of 1.0 mm in diameter [21]. The cooking proceeded by immersing MBC in a beaker with boiling water (98 °C) over a hotplate. The 50% cooked point, indicated by plungers dropping and penetrating 13 (approximately 50% of the 25 individual seeds) of the individual beans, corresponds to the sensory preferred degree of cooking, according to methodology adapted from Proctor and Watts [11,22]. A digital chronometer was used to record the cooking time during the process.

### 2.5. Yield per Plant Data Collection and Evaluation

The yield per plant parameter was evaluated by the method adopted from Adewale [23] and converted to the unit used by Bisht [17]. A total of 10 seeds of each accession were planted in eight blocks of 26 lines as earlier described in Section 2.2. Matured pods from the 10 plants of each accession were harvested, threshed, sun-dried, and weighed. The weight of total seeds for each plant was then recorded and the mean seed weight for all the plants harvested on a line (plot) was evaluated as yield per plant. Similarly, the mean seed weight for all the accessions of the same species was evaluated as the yield per plant for the species.

### 2.6. Data Analysis

The values for water absorption capacity and cooking time were recorded in triplicate and presented as mean ± standard error using XLSTAT. The data were subjected to two-way analysis of variance (ANOVA), correlation coefficients, and Tukey's test. $p < 0.05$ was considered statistically significant. Agglomerative Hierarchical Clustering (AHC) analysis was performed to examine similarities between accessions. Descriptive statistics for the yield traits as well as cooking time and water absorption capacity were also computed using XLSTAT. All the data were entered in an excel sheet and analyzed using XLSTAT-Base version 21.1.57988.0.

## 3. Results

### 3.1. Cooking Time and Water Absorption Capacity of Domesticated Legumes

The water absorptions and the cooking time for a landrace of *Vigna vexillata* (check 1), cowpea (check 2), and rice bean (check 3) used here were harvested from two different agro-ecological zones, as shown in Table 2a.

The values for both water absorption and cooking time showed no significant difference between agroecological zones and between the three species and therefore no environment × species interaction (Table 2a). A detailed presentation of the interactions between species (*V. vexillata* landrace, *V. unguiculata*, and *V. umbellate*) as replicated within locations is shown in Table 2b,c. It shows that there is no replicate interaction effect between species within locations for the water absorption capacity trait in all the tested combinations. However, replicate interaction effects were significant ($p < 0.05$) when tested within locations between species for the cooking time trait except when tested across locations (Table 2c).

The values for cooking time showed significant differences between the three domesticated varieties ($p < 0.05$). Pearson correlation analysis shows that there is no correlation between the water absorption capacity and cooking time considering only the three seed varieties ($r = -0.030$ for site A, 0.029 for site B) (Figure 2). Cowpea has a higher cooking time than rice bean which also cook longer than the landrace of *V. vexillata*.

**Table 2.** Results of the cooking time and water absorption capacity for the domesticated legume seeds. (**a**) Means, analysis of variance and type III sum of square analysis for the cooking time and water absorption capacity traits of domesticated legume seeds. (**b**) Details of interactions within locations effects for water absorption capacity trait. (**c**) Details of interactions within locations effects for cooking time trait.

**(a)**

| | Water Absorption Capacity | | Cooking Time (min) | |
|---|---|---|---|---|
| **Checks** | **Site A** | **Site B** | **Site A** | **Site B** |
| Landrace of *Vigna vexillata* | 1.33 ± 0.11 [a] | 1.32 ± 0.13 [a] | 10.24 ± 0.15 [a] | 10.26 ± 0.15 [a] |
| Cowpea (*Vigna unguiculata*) | 1.27 ± 0.08 [a] | 1.27 ± 0.08 [a] | 16.29 ± 0.15 [c] | 16.31 ± 0.15 [c] |
| Rice Bean (*Vigna umbellata*) | 1.16 ± 0.06 [a] | 1.16 ± 0.06 [a] | 13.20 ± 0.12 [b] | 13.23 ± 0.12 [b] |

**Analysis of Variance (ANOVA)**

| | Water Absorption Capacity | | | | | Cooking Time (min) | | | | |
|---|---|---|---|---|---|---|---|---|---|---|
| **Source** | **DF** | **Sum of Squares** | **Mean Squares** | **F** | **p** | **DF** | **Sum of Squares** | **Mean Squares** | **F** | **p** |
| Model | 5 | 1.263 | 0.253 | 1.134 | 0.343 | 5 | 1582.515 | 316.503 | 356.710 | <0.0001 |
| Error | 258 | 57.475 | 0.223 | | | 258 | 228.919 | 0.887 | | |
| Corrected Total | 263 | 58.738 | | | | 263 | 1811.434 | | | |

**Type III Sum of Squares Analysis**

| **Source** | **DF** | **Sum of Squares** | **Mean Squares** | **F** | **p** | **DF** | **Sum of Squares** | **Mean Squares** | **F** | **p** |
|---|---|---|---|---|---|---|---|---|---|---|
| Location (Site) Effect | 1 | 0.001 | 0.001 | 0.004 | 0.950 | 1 | 0.044 | 0.044 | 0.050 | 0.823 |
| Species Effect | 2 | 1.262 | 0.631 | 2.833 | 0.061 | 2 | 1582.470 | 791.235 | 891.749 | <0.0001 |
| Location X Species | 2 | 0.000 | 0.000 | 0.000 | 1.000 | 2 | 0.000 | 0.000 | 0.000 | 1.000 |

Results are represented as the mean value of triplicates ± standard error. Different letters in the same column represent statistically different mean values (*p* = 0.05). Site A: TARI-Selian; Site B: TaCRI. *DF*: Degree of freedom; *F*: *F*-ratio; *p*: *p*-value.

(**b**)

| Location × Species/Tukey (HSD)/Analysis of the Differences between the Categories with a Confidence Interval of 95% (Water Absorption Capacity) | | | | | |
|---|---|---|---|---|---|
| **Contrast** | **Difference** | **Standardized Difference** | **Critical value** | **Pr > Diff** | **Significant** |
| Location-Site A × Species-Check 1 vs. Location-Site B × Species-Check 3 | 0.173 | 1.675 | 2.871 | 0.550 | No |
| Location-Site A × Species-Check 1 vs. Location-Site A × Species-Check 3 | 0.172 | 1.661 | 2.871 | 0.559 | No |
| Location-Site A × Species-Check 1 vs. Location-Site B × Species-Check 2 | 0.063 | 0.620 | 2.871 | 0.990 | No |
| Location-Site A × Species-Check 1 vs. Location-Site A × Species-Check 2 | 0.059 | 0.584 | 2.871 | 0.992 | No |
| Location-Site A × Species-Check 1 vs. Location-Site B × Species-Check 1 | 0.006 | 0.054 | 2.871 | 1.000 | No |
| Location-Site B × Species-Check 1 vs. Location-Site B × Species-Check 3 | 0.167 | 1.619 | 2.871 | 0.587 | No |
| Location-Site B × Species-Check 1 vs. Location-Site A × Species-Check 3 | 0.166 | 1.605 | 2.871 | 0.596 | No |
| Location-Site B × Species-Check 1 vs. Location-Site B × Species-Check 2 | 0.057 | 0.563 | 2.871 | 0.993 | No |
| Location-Site B × Species-Check 1 vs. Location-Site A × Species-Check 2 | 0.054 | 0.527 | 2.871 | 0.995 | No |
| Location-Site A × Species-Check 2 vs. Location-Site B × Species-Check 3 | 0.114 | 1.160 | 2.871 | 0.855 | No |
| Location-Site A × Species-Check 2 vs. Location-Site A × Species-Check 3 | 0.112 | 1.145 | 2.871 | 0.862 | No |
| Location-Site A × Species-Check 2 vs. Location-Site B × Species-Check 2 | 0.004 | 0.038 | 2.871 | 1.000 | No |
| Location-Site B × Species-Check 2 vs. Location-Site B × Species-Check 3 | 0.110 | 1.122 | 2.871 | 0.872 | No |
| Location-Site B × Species-Check 2 vs. Location-Site A × Species-Check 3 | 0.108 | 1.107 | 2.871 | 0.878 | No |
| Location-Site A × Species-Check 3 vs. Location-Site B × Species-Check 3 | 0.001 | 0.014 | 2.871 | 1.000 | No |
| Tukey's d critical value | | | 4.061 | | |

Check 1: Landrace of *Vigna* vexillata; Check 2: Cowpea (*Vigna unguiculata*); Check 3: Rice Bean (*Vigna umbellata*).

(**c**)

| Location × Species/Tukey (HSD)/Analysis of the Differences between the Categories with a Confidence Interval of 95% (Cooking Time) | | | | | |
|---|---|---|---|---|---|
| *Contrast* | Difference | Standardized Difference | Critical value | Pr > Diff | Significant |
| Location-Site B × Species-Check 2 vs. Location-Site A × Species-Check 1 | 6.074 | 29.913 | 2.871 | <0.0001 | Yes |
| Location-Site B × Species-Check 2 vs. Location-Site B × Species-Check 1 | 6.048 | 29.785 | 2.871 | <0.0001 | Yes |
| Location-Site B × Species-Check 2 vs. Location-Site A × Species-Check 3 | 3.109 | 15.908 | 2.871 | <0.0001 | Yes |
| Location-Site B × Species-Check 2 vs. Location-Site B × Species-Check 3 | 3.083 | 15.775 | 2.871 | <0.0001 | Yes |
| Location-Site B × Species-Check 2 vs. Location-Site A × Species-Check 2 | 0.026 | 0.135 | 2.871 | 1.000 | No |
| Location-Site A × Species-Check 2 vs. Location-Site A × Species-Check 1 | 6.048 | 29.785 | 2.871 | <0.0001 | Yes |
| Location-Site A × Species-Check 2 vs. Location-Site B × Species-Check 1 | 6.022 | 29.657 | 2.871 | <0.0001 | Yes |
| Location-Site A × Species-Check 2 vs. Location-Site A × Species-Check 3 | 3.083 | 15.775 | 2.871 | <0.0001 | Yes |
| Location-Site A × Species-Check 2 vs. Location-Site B × Species-Check 3 | 3.057 | 15.642 | 2.871 | <0.0001 | Yes |
| Location-Site B × Species-Check 3 vs. Location-Site A × Species-Check 1 | 2.991 | 14.514 | 2.871 | <0.0001 | Yes |
| Location-Site B × Species-Check 3 vs. Location-Site B × Species-Check 1 | 2.965 | 14.388 | 2.871 | <0.0001 | Yes |
| Location-Site B × Species-Check 3 vs. Location-Site A × Species-Check 3 | 0.026 | 0.131 | 2.871 | 1.000 | No |
| Location-Site A × Species-Check 3 vs. Location-Site A × Species-Check 1 | 2.965 | 14.388 | 2.871 | <0.0001 | Yes |
| Location-Site A × Species-Check 3 vs. Location-Site B × Species-Check 1 | 2.939 | 14.262 | 2.871 | <0.0001 | Yes |
| Location-Site B × Species-Check 1 vs. Location-Site A × Species-Check 1 | 0.026 | 0.122 | 2.871 | 1.000 | No |
| Tukey's d critical value | | | 4.061 | | |

Check 1: Landrace of *Vigna vexillata*; Check 2: Cowpea (*Vigna unguiculata*); Check 3: Rice Bean (*Vigna umbellata*).

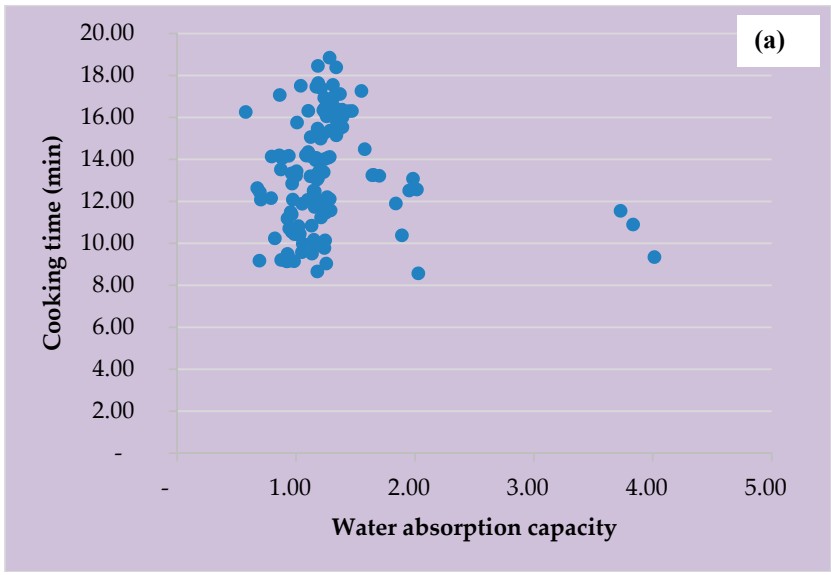

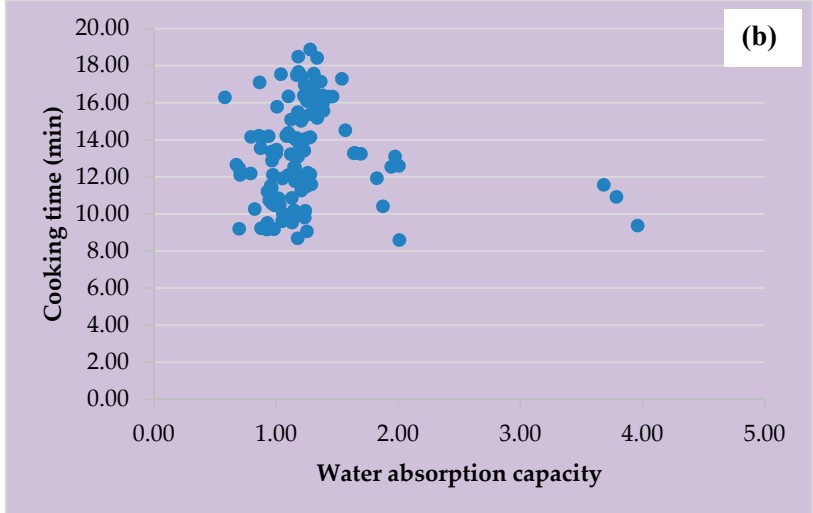

**Figure 2.** Correlation between water absorption and cooking time for the three checks. (**a**) Plotted with data from Site A; (**b**) plotted with data from Site B.

### 3.2. Cooking Time and Water Absorption Capacity of Vigna ambacensis Accessions

The water absorption capacities and the cooking times for 11 accessions of wild *Vigna ambacensis* are presented in Table 3.

The values for water absorption capacity and cooking time showed no significant difference ($p > 0.05$) when compared with the values of their corresponding accession harvested in the other agro-ecological zone (Table 3).

Considering the water absorption capacity, all the wild accessions exhibited significantly low values as compared with all three checks. The water absorption capacity of wild accessions varied from 0.08 ± 0.01 to 0.47 ± 0.01 (Table 3) in both site A and B. Accession TVNu342 showed no significant difference in water absorption capacity with three checks and with accession TVNu219.

The cooking time of the wild accessions varied from 23.02 ± 0.50 to 24.26 ± 0.07 min in both sites (Table 3). All the wild accessions possessed significantly higher cooking time values compared with the three checks (Table 3). None of the accessions cooked faster than the checks.

**Table 3.** Cooking time and water absorption of *Vigna ambacensis* accessions.

| Species/Accession Number | Water Absorption Capacity | | Cooking Time (min) | |
|---|---|---|---|---|
| | Site A | Site B | Site A | Site B |
| Landrace of *Vigna vexillata* | 1.33 ± 0.11 [a] | 1.32 ± 0.13 [a] | 10.24 ± 0.15 [a] | 10.26 ± 0.15 [a] |
| Cowpea (*Vigna unguiculata*) | 1.27 ± 0.08 [a] | 1.27 ± 0.08 [a] | 16.29 ± 0.15 [b] | 16.31 ± 0.15 [b] |
| Rice bean (*Vigna umbellata*) | 1.16 ± 0.06 [a] | 1.16 ± 0.06 [a] | 13.20 ± 0.12 [c] | 13.23 ± 0.12 [c] |
| TVNu1699 | 0.14 ± 0.01 [c] | 0.13 ± 0.01 [c] | 24.26 ± 0.07 [d] | 23.87 ± 0.10 [d] |
| TVNu342 | 0.47 ± 0.01 [a,b] | 0.45 ± 0.01 [a,b] | 23.34 ± 0.16 [d] | 23.35 ± 0.18 [d] |
| TVNu877 | 0.22 ± 0.01[c] | 0.21 ± 0.01 [c] | 24.10 ± 0.19 [d] | 23.71 ± 0.22 [d] |
| TVNu223 | 0.21 ± 0.01 [c] | 0.21 ± 0.02 [c] | 23.35 ± 0.55 [d] | 23.36 ± 0.50 [d] |
| TVNu720 | 0.22 ± 0.01 [c] | 0.20 ± 0.01 [c] | 23.02 ± 0.50 [d] | 23.03 ± 0.45 [d] |
| TVNu219 | 0.28 ± 0.02 [b,c] | 0.26 ± 0.01 [b,c] | 24.06 ± 0.49 [d] | 24.08 ± 0.50 [d] |
| TVNu1840 | 0.11 ± 0.01 [c] | 0.10 ± 0.01 [c] | 23.36 ± 0.21 [d] | 23.37 ± 0.30 [d] |
| TVNu1804 | 0.09 ± 0.01 [c] | 0.08 ± 0.01 [c] | 23.55 ± 0.52 [d] | 23.56 ± 0.50 [d] |
| TVNu1792 | 0.23 ± 0.01 [c] | 0.09 ± 0.01 [c] | 23.28 ± 0.22 [d] | 23.30 ± 0.30 [d] |
| TVNu1644 | 0.09 ± 0.01 [c] | 0.21 ± 0.01 [c] | 23.12 ± 0.10 [d] | 23.13 ± 0.15 [d] |
| TVNu1185 | 0.12 ± 0.01 [c] | 0.11 ± 0.01 [c] | 23.34 ± 0.33 [d] | 23.35 ± 0.30 [d] |

**Analysis of Variance (ANOVA)**

| | Water Absorption Capacity | | | | | Cooking Time (min) | | | | |
|---|---|---|---|---|---|---|---|---|---|---|
| Source | DF | Sum of Squares | Mean Squares | F | p | DF | Sum of Squares | Mean Squares | F | p |
| Model | 27 | 60.707 | 2.248 | 11.756 | <0.0001 | 27 | 6864.480 | 254.240 | 313.317 | <0.0001 |
| Error | 302 | 57.761 | 0.191 | | | 302 | 245.057 | 0.811 | | |
| Corrected Total | 329 | 118.469 | | | | 329 | 7109.537 | | | |

**Type III Sum of Squares Analysis**

| Source | DF | Sum of squares | Mean squares | F | p | DF | Sum of Squares | Mean Squares | F | p |
|---|---|---|---|---|---|---|---|---|---|---|
| Location (Site) | 1 | 0.002 | 0.002 | 0.011 | 0.916 | 1 | 0.007 | 0.007 | 0.008 | 0.929 |
| Genotype (Accessions) | 13 | 60.704 | 4.670 | 24.414 | <0.0001 | 13 | 6864.433 | 528.033 | 650.730 | <0.0001 |
| Location × Genotype | 13 | 0.001 | 0.000 | 0.001 | 1.000 | 13 | 0.002 | 0.000 | 0.000 | 1.000 |

Results are represented as the mean value of triplicates ± standard error. Mean values without any letter in common within each column are significantly different ($p$ = 0.05). Site A: TARI-Selian; Site B: TaCRI. *DF*: Degree of freedom; *F*: F-ratio; *p*: *p*-value.

Additionally, there was no correlation between the water absorption capacity and cooking time considering only the 11 accessions studied ($r = -0.025$ for site A and $r = -0.024$ for site B) (Figure 3).

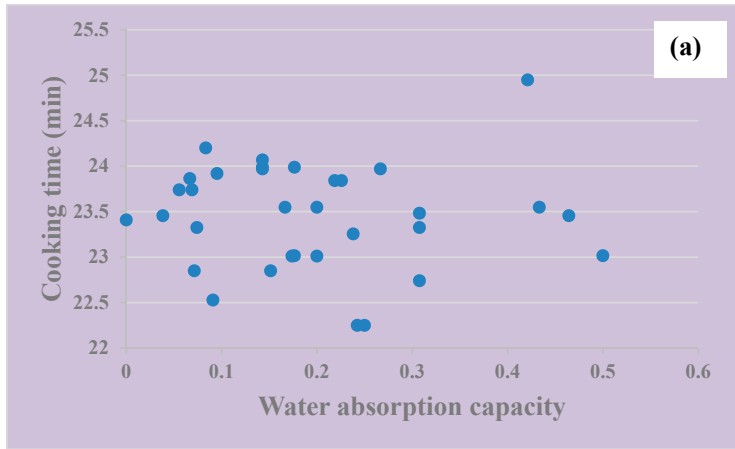

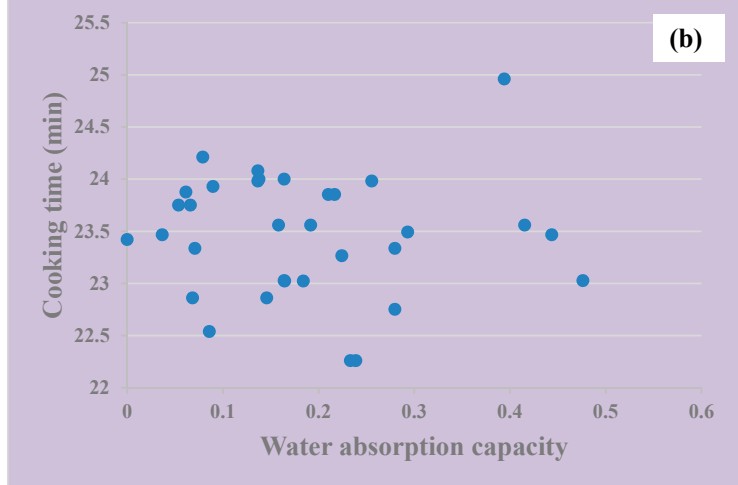

**Figure 3.** Correlation between water absorption and cooking time for the *Vigna ambacensis* accessions. (**a**) Plotted with data from Site A; (**b**) plotted with data from Site B.

### 3.3. Cooking Time and Water Absorption Capacity of Vigna vexillata Accessions

The result for water absorption capacity and cooking time for 35 accessions of wild *Vigna vexillata* is shown in Table 4. The values for water absorption capacity and cooking time show no significant difference ($p > 0.05$) when compared with the values of their corresponding accessions harvested in the other agro-ecological zone.

The Water Absorption Capacity in all the wild accessions with exception of TVNu781 and TVNu837 showed significant low values compared with the three checks (Table 4). The water absorption capacity of the wild *V. vexillata* accessions varied from $0.04 \pm 0.00$ to $1.10 \pm 0.03$ in both site A and B (Table 4).

Considering the cooking time, there is a high diversity in differences among the accessions. The cooking time varied from $16.22 \pm 0.23$ to $31.04 \pm 0.33$ min in site A and from $16.24 \pm 0.20$ to $31.06 \pm 0.31$ min in site B (Table 4). Accessions TVNu781, AGG308107WVIG2, AGG308097WVIG1, and TVNU1624 exhibited relatively similar cooking time with check 2 (Cowpea) (Table 4). Conversely, cooking time for all other remaining accessions was significantly higher than all the checks. Pearson correlation analysis shows that there is a weak negative correlation between the water absorption capacity and cooking time considering the wild *V. vexillata* tested ($r = -0.31$ for site A and $r = -0.32$). Furthermore, the regression analysis shows that the water absorption capacity and cooking time are related by the equation: $Y = -5.12x + 27.15$ with $R^2 = 0.094$ (Figure 4).

**Table 4.** Cooking time and water absorption capacity of *Vigna vexillata* accessions.

| Species/Accession Number | Water Absorption Capacity | | Cooking Time (min) | |
|---|---|---|---|---|
| | Site A | Site B | Site A | Site B |
| Landrace of *Vigna vexillata* | 1.33 ± 0.11 [a] | 1.32 ± 0.13 [a] | 10.24 ± 0.15 [n] | 10.26 ± 0.15 [n] |
| Cowpea (*Vigna unguiculata*) | 1.27 ± 0.08 [a,b,c] | 1.27 ± 0.08 [a,b,c] | 16.29 ± 0.15 [l] | 16.31 ± 0.15 [l] |
| Rice Bean (*Vigna umbellata*) | 1.16 ± 0.06 [a,b,c] | 1.16 ± 0.06 [a,b,c] | 13.20 ± 0.12 [m] | 13.23 ± 0.12 [m] |
| TVNu781 | 1.10 ± 0.02 [abcd] | 1.10 ± 0.01 [abcd] | 31.04 ± 0.33 [a,b] | 31.06 ± 0.31 [a,b] |
| TVNu837 | 1.07 ± 0.01 [abcd] | 1.05 ± 0.01 [abcd] | 29.34 ± 0.32 [a,b,c,d,e,f,g] | 29.35 ± 0.01 [a,b,c,d,e,f,g] |
| TVNu1582 | 0.73 ± 0.01 [abcd] | 0.67 ± 0.02 [abcd] | 16.25 ± 0.24 [l] | 16.26 ± 0.30 [l] |
| TVNu1358 | 0.57 ± 0.01 [abcd] | 0.53 ± 0.01 [abcd] | 17.37 ± 0.26 [l] | 17.38 ± 0.28 [l] |
| AGG308107WVIG2 | 0.43 ± 0.01 [abcd] | 0.41 ± 0.01 [abcd] | 26.32 ± 0.49 [f,g,h,i,j] | 26.33 ± 0.51 [f,g,h,i,j] |
| TVNu1593 | 0.42 ± 0.01 [abcd] | 0.38 ± 0.01 [bcd] | 16.22 ± 0.23 [l] | 16.24 ± 0.20 [l] |
| TVNu1591 | 0.41 ± 0.01 [abcd] | 0.38 ± 0.01 [bcd] | 26.38 ± 0.40 [f,g,h,i,j] | 26.39 ± 0.43 [f,g,h,i,j] |
| TVNu120 | 0.40 ± 0.01 [abcd] | 0.38 ± 0.01 [bcd] | 31.10 ± 0.31 [a] | 30.71 ± 0.34 [a] |
| TVNu333 | 0.40 ± 0.02 [abcd] | 0.37 ± 0.02 [bcd] | 26.28 ± 0.40 [f,g,h,i,j] | 26.30 ± 0.35 [f,g,h,i,j] |
| TVNu1546 | 0.39 ± 0.02 [bcd] | 0.37 ± 0.01 [bcd] | 29.07 ± 0.13 [a,b,c,d,e,f] | 29.08 ± 0.15 [a,b,c,d,e,f] |
| AGG308101WVIG1 | 0.37 ± 0.01 [bcd] | 0.34 ± 0.01 [bcd] | 29.36 ± 0.50 [a,b,c,d,e] | 29.37 ± 0.47 [a,b,c,d,e] |
| TVNu1701 | 0.35 ± 0.01 [bcd] | 0.33 ± 0.01 [cd] | 24.59 ± 0.50 [j] | 24.60 ± 0.57 [j] |
| AGG308096 WVIG2 | 0.34 ± 0.01 [cd] | 0.32 ± 0.01 [cd] | 26.47 ± 0.59 [f,g,h,i,j] | 26.49 ± 0.60 [f,g,h,i,j] |
| TVNu1629 | 0.33 ± 0.01 [cd] | 0.32 ± 0.02 [cd] | 28.19 ± 1.15 [b,c,d,e,f,g,h,i] | 28.20 ± 1.20 [b,c,d,e,f,g,h,i] |
| TVNu293 | 0.33 ± 0.01 [cd] | 0.31 ± 0.01 [cd] | 26.32 ± 0.28 [f,g,h,i,j] | 26.33 ± 0.30 [f,g,h,i,j] |
| TVNu832 | 0.32 ± 0.01 [cd] | 0.30 ± 0.01 [cd] | 25.46 ± 0.36 [h,i,j] | 25.47 ± 0.36 [h,i,j] |
| TVNu1796 | 0.32 ± 0.01 [cd] | 0.30 ± 0.01 [cd] | 27.38 ± 0.48 [c,d,e,f,g,h,i,j] | 27.39 ± 0.50 [c,d,e,f,g,h,i,j] |
| TVNu1529 | 0.32 ± 0.01 [cd] | 0.30 ± 0.01 [cd] | 27.29 ± 0.64 [c,d,e,f,g,h,i,j] | 27.30 ± 0.64 [c,d,e,f,g,h,i,j] |
| TVNu1628 | 0.30 ± 0.01 [cd] | 0.28 ± 0.01 [cd] | 26.30 ± 0.36 [f,g,h,i,j] | 26.31 ± 0.33 [f,g,h,i,j] |
| TVNu1344 | 0.29 ± 0.01 [cd] | 0.28 ± 0.01 [cd] | 30.03 ± 0.44 [a,b,c] | 30.04 ± 0.44 [a,b,c] |
| TVNu1632 | 0.29 ± 0.01 [cd] | 0.28 ± 0.01 [cd] | 29.41 ± 0.52 [a,b,c,d] | 28.25 ± 0.50 [a,b,c,d] |
| TVNu1370 | 0.28 ± 0.01 [cd] | 0.26 ± 0.02 [cd] | 28.23 ± 0.39 [l] | 29.43 ± 0.40 [l] |
| TVNu1360 | 0.28 ± 0.01 [cd] | 0.25 ± 0.01 [cd] | 27.05 ± 0.71 [d,e,f,g,h,i,j] | 27.60 ± 0.72 [d,e,f,g,h,i,j] |
| TVNu1624 | 0.25 ± 0.01 [cd] | 0.23 ± 0.01 [d] | 26.28 ± 0.46 [f,g,h,i,j] | 26.29 ± 0.46 [f,g,h,i,j] |
| TVNu1621 | 0.25 ± 0.01 [cd] | 0.23 ± 0.01 [d] | 17.24 ± 0.47 [l] | 17.26 ± 0.48 [l] |
| AGG62154WVIG_1 | 0.20 ± 0.01 [d] | 0.19 ± 0.01 [d] | 21.33 ± 0.17 [k] | 21.34 ± 0.17 [k] |
| TVNu1092 | 0.19 ± 0.01 [d] | 0.18 ± 0.01 [d] | 29.02 ± 0.23 [a,b,c,d,e,f,g] | 29.04 ± 0.55 [a,b,c,d,e,f,g] |
| TVNu479 | 0.18 ± 0.01 [d] | 0.17 ± 0.01 [d] | 26.50 ± 0.56 [e,f,g,h,i,j] | 26.52 ± 0.20 [e,f,g,h,i,j] |
| AGG308097WVIG 1 | 0.17 ± 0.01 [d] | 0.16 ± 0.01 [d] | 28.56 ± 0.50 [a,b,c,d,e,f,g] | 28.58 ± 0.50 [a,b,c,d,e,f,g] |
| TVNu178 | 0.17 ± 0.01 [d] | 0.16 ± 0.01 [d] | 17.03 ± 0.54 [l] | 16.64 ± 0.01 [l] |
| TVNu955 | 0.11 ± 0.01 [d] | 0.11 ± 0.01 [d] | 25.28 ± 0.47 [i,j] | 25.29 ± 0.47 [i,j] |
| TVNu1378 | 0.11 ± 0.01 [d] | 0.10 ± 0.00 [d] | 16.29 ± 0.45 [l] | 16.31 ± 0.47 [l] |
| TVNu1586 | 0.06 ± 0.00 [d] | 0.05 ± 0.01 [d] | 28.39 ± 0.29 [a,b,c,d,e,f,g] | 28.41 ± 0.30 [a,b,c,d,e,f,g] |
| TVNu381 | 0.04 ± 0.00 [d] | 0.042 ± 0.00 [d] | 25.41 ± 0.63 [h,i,j] | 25.42 ± 0.64 [h,i,j] |
| AGG308099WVIG2 | 0.042 ± 0.01 [d] | 0.04 ± 0.01 [d] | 26.16 ± 0.48 [h,i,j] | 26.17 ± 0.50 [h,i,j] |

**Table 4.** *Cont.*

| Species/Accession Number | Water Absorption Capacity | | | Cooking Time (min) | | |
|---|---|---|---|---|---|---|
| | Site A | Site B | | Site A | Site B | |
| Analysis of Variance (ANOVA) | | | | | | |
| | Water Absorption Capacity | | | | Cooking Time (min) | |

| Source | DF | Sum of Squares | Mean Squares | F | p | DF | Sum of Squares | Mean Squares | F | p |
|---|---|---|---|---|---|---|---|---|---|---|
| Model | 75 | 111.003 | 1.480 | 9.649 | <0.0001 | 75 | 22,437.582 | 299.168 | 368.513 | <0.0001 |
| Error | 398 | 61.050 | 0.153 | | | 398 | 323.106 | 0.812 | | |
| Corrected Total | 473 | 172.052 | | | | 473 | 22,760.688 | | | |

| Type III Sum of Squares Analysis | | | | | | | | | | |
|---|---|---|---|---|---|---|---|---|---|---|
| Source | DF | Sum of Squares | Mean Squares | F | p | DF | Sum of Squares | Mean Squares | F | p |
| Location (Site) | 1 | 0.018 | 0.018 | 0.117 | 0.732 | 1 | 0.012 | 0.012 | 0.015 | 0.903 |
| Genotype (Accessions) | 37 | 110.978 | 2.999 | 19.554 | <0.0001 | 37 | 22,437.529 | 606.420 | 746.983 | <0.0001 |
| Location × Genotype | 37 | 0.013 | 0.000 | 0.002 | 1.000 | 37 | 0.005 | 0.000 | 0.000 | 1.000 |

Results are represented as the mean value of triplicates ± standard error. Mean values without any letter in common within each column are significantly different ($p = 0.05$). Site A: TARI-Selian; Site B: TaCRI. *DF*: Degree of freedom; *F*: *F*-ratio; *p*: *p*-value.

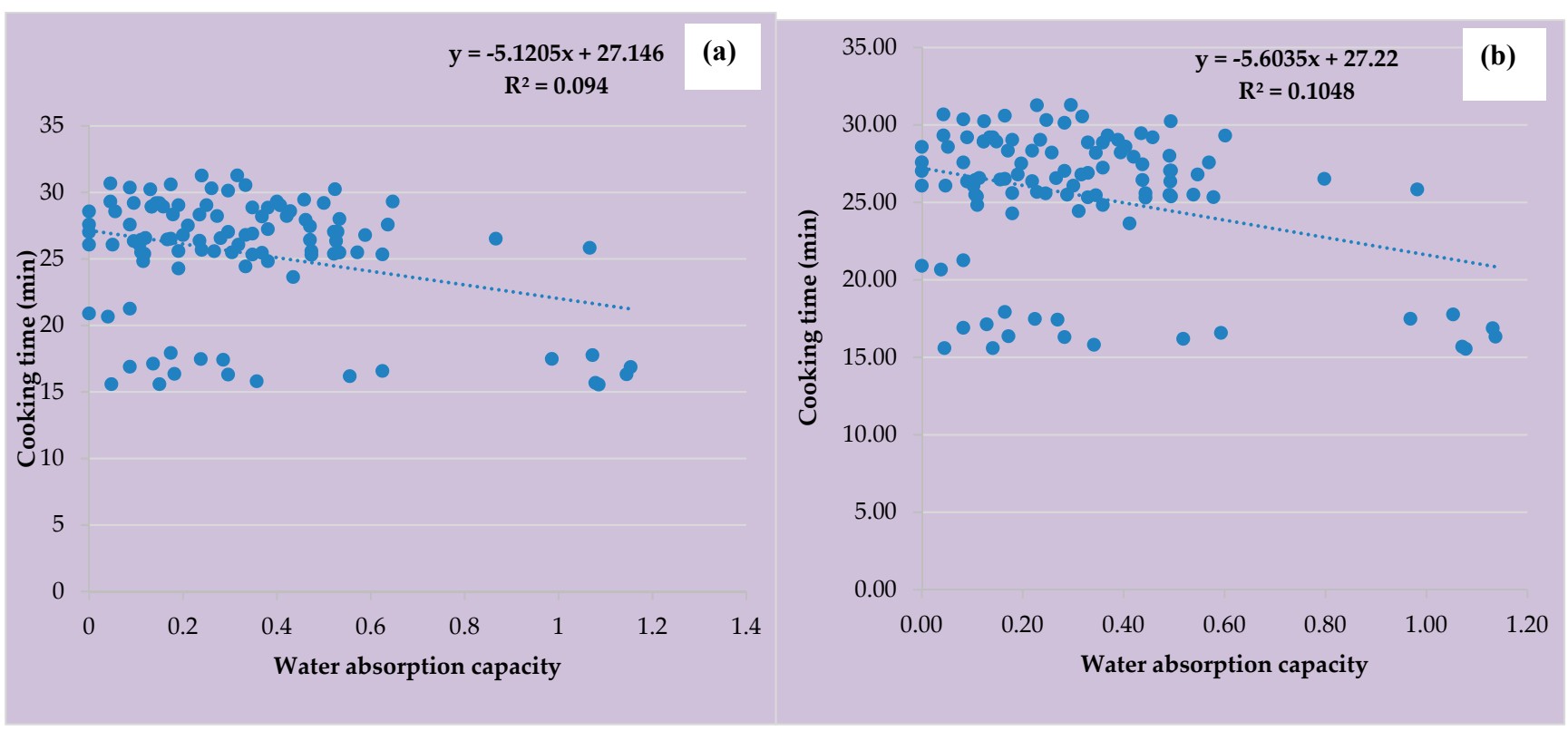

**Figure 4.** Correlation between water absorption and cooking time for the *Vigna vexillata* accessions. (**a**) Plotted with data from Site A; (**b**) plotted with data from Site B.

### 3.4. Cooking Time and Water Absorption capacity of Vigna reticulata Accessions

Table 5 shows the various values for water absorption capacity and cooking time for 32 accessions of wild *Vigna reticulata*. The values for water absorption capacity and cooking time showed no significant difference ($p > 0.05$) when compared with the values of their corresponding accessions harvested in the other agro-ecological zone.

All the wild accessions showed significantly low water absorption capacity values compared with the checks except for TVNu1520, and TVNu325 (Table 5). The water absorption capacity of the wild *V. reticulata* accessions varied from $0.06 \pm 0.01$ to $1.27 \pm 0.08$ in site A and from $0.06 \pm 0.01$ to $1.32 \pm 0.13$ in site B (Table 5). No location and genotype × location interactions ($p > 0.05$) were observed for both water absorption capacity and cooking time traits in these accessions. However, only genotype interaction was observed for both traits ($p < 0.05$).

Regarding cooking time, there is a high diversity in differences of means among the accessions. Twenty-five accessions showed significant higher cooking time values. Check 2 showed no significant difference in cooking time with TVNu325 and the unknown *V. reticulata* accession only. The cooking times for all accessions varied from $17.41 \pm 0.44$ to $30.25 \pm 0.41$ min in site A and from $17.42 \pm 0.45$ to $30.26 \pm 0.42$ min in site B (Table 5).

Pearson correlation analysis shows that there is a weak negative correlation between the water absorption and cooking time considering the wild *V. reticulata* tested ($r = -0.43$ for site A and $r = -0.45$) (Figure 5. Furthermore, the regression analysis shows that the water absorption and cooking time are related by the equation: $Y = -2.57x + 27.77$ with $R^2 = 0.18$ (Figure 5).

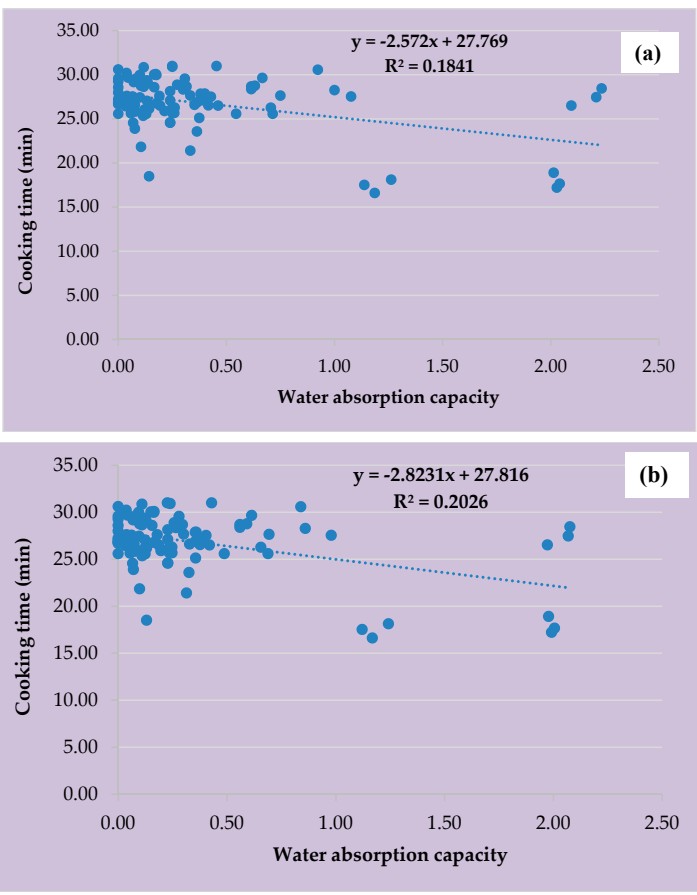

**Figure 5.** Correlation between water absorption and cooking time for the *Vigna reticulata* accessions. (**a**) Plotted with data from Site A; (**b**) plotted with data from Site B.

**Table 5.** Cooking time and water absorption capacity of *Vigna reticulata* accessions.

| Species/Accession Number | Water Absorption Capacity | | Cooking Time (min) | |
|---|---|---|---|---|
| | Site A | Site B | Site A | Site B |
| Landrace of *Vigna vexillata* | 1.33 ± 0.11 [a,b,c] | 1.32 ± 0.13 [a,b,c] | 10.24 ± 0.15 [h] | 10.26 ± 0.15 [h] |
| Cowpea (*Vigna unguiculata*) | 1.27 ± 0.08 [a,b,c,d] | 1.27 ± 0.08 [a,b,c,d] | 16.29 ± 0.15 [f] | 16.31 ± 0.15 [f] |
| Rice Bean (*Vigna umbellata*) | 1.16 ± 0.06 [a,b,c,d] | 1.16 ± 0.06 [a,b,c,d] | 13.20 ± 0.12 [g] | 13.23 ± 0.12 [g] |
| TVNu324 | 0.49 ± 0.02 [c,d] | 0.47 ± 0.02 [c,d] | 26.51 ± 0.47 [a,b,c,d] | 26.53 ± 0.48 [a,b,c,d] |
| TVNu325 | 2.03 ± 0.02 [a,b] | 1.99 ± 0.01 [a,b] | 17.92 ± 0.51 [f] | 17.93 ± 0.52 [f] |
| Unknown _*Vigna reticulata* | 1.20 ± 0.02 [a,b,c,d] | 1.18 ± 0.02 [a,b,c,d] | 17.41 ± 0.44 [f] | 17.42 ± 0.45 [f] |
| TVNu343 | 0.19 ± 0.01 [c,d] | 0.18 ± 0.01 [c,d] | 30.25 ± 0.41 [a] | 30.26 ± 0.42 [a] |
| TVNu767 | 0.12 ± 0.01 [c,d] | 0.11 ± 0.01 [c,d] | 29.18 ± 0.99 [a,b,c,d] | 29.20 ± 1.00 [a,b,c,d] |
| TVNu1520 | 2.18 ± 0.03 [a] | 2.04 ± 0.03 [a] | 27.46 ± 0.91 [a,b,c,d] | 27.48 ± 0.92 [a,b,c,d] |
| TVNu349 | 0.31 ± 0.02 [c,d] | 0.29 ± 0.01 [c,d] | 29.14 ± 0.74 [a,b,c,d] | 28.76 ± 0.75 [a,b,c,d] |
| TVNu379 | 0.77 ± 0.01 [c,d] | 0.71 ± 0.02 [c,d] | 29.38 ± 0.46 [a,b,c,d] | 28.99 ± 0.44 [a,b,c,d] |
| TVNu524 | 0.17 ± 0.01 [c,d] | 0.17 ± 0.01 [c,d] | 25.57 ± 0.57 [c,d,e] | 25.58 ± 0.58 [c,d,e] |
| TVNu1698 | 0.12 ± 0.01 [c,d] | 0.11 ± 0.01 [c,d] | 26.34 ± 0.56 [b,c,d,e] | 26.36 ± 0.57 [b,c,d,e] |
| TVNu1191 | 0.22 ± 0.01 [c,d] | 0.21 ± 0.01 [c,d] | 25.38 ± 1.00 [d,e] | 25.39 ± 0.99 [d,e] |
| TVNu1394 | 0.82 ± 0.02 [b,c,d] | 0.75 ± 0.01 [b,c,d] | 28.21 ± 0.99 [a,b,c,d] | 27.82 ± 0.97 [a,b,c,d] |
| TVNu-224 | 0.19 ± 0.01 [c,d] | 0.18 ± 0.01 [c,d] | 25.55 ± 0.51 [c,d,e] | 25.57 ± 0.52 [c,d,e] |
| TVNu739 | 0.15 ± 0.01 [c,d] | 0.14 ± 0.01 [c,d] | 28.50 ± 0.46 [a,b,c,d] | 28.52 ± 0.47 [a,b,c,d] |
| TVNu56 | 0.24 ± 0.02 [c,d] | 0.22 ± 0.02 [c,d] | 27.01 ± 2.73 [a,b,c,d] | 26.62 ± 2.70 [a,b,c,d] |
| TVNu1405 | 0.29 ± 0.02 [c,d] | 0.26 ± 0.02 [c,d] | 30.03 ± 0.64 [a,b] | 29.64 ± 0.62 [a,b] |
| TVNu607 | 0.08 ± 0.01 [d] | 0.08 ± 0.01 [d] | 27.33 ± 0.49 [a,b,c,d] | 26.38 ± 0.47 [a,b,c,d] |
| TVNu916 | 0.12 ± 0.01 [c,d] | 0.11 ± 0.01 [c,d] | 26.37 ± 0.52 [b,c,d,e] | 27.84 ± 0.55 [b,c,d,e] |
| AGG17856WVIG 1 | 0.16 ± 0.01 [c,d] | 0.15 ± 0.01 [c,d] | 28.23 ± 1.00 [a,b,c,d] | 27.35 ± 0.97 [a,b,c,d] |
| TVNu1790 | 0.32 ± 0.02 [c,d] | 0.29 ± 0.02 [c,d] | 28.43 ± 0.47 [a,b,c,d] | 28.44 ± 0.47 [a,b,c,d] |
| TVNu491 | 0.15 ± 0.01 [c,d] | 0.14 ± 0.01 [c,d] | 28.44 ± 0.93 [a,b,c,d] | 28.45 ± 0.92 [a,b,c,d] |
| TVNu1808 | 0.16 ± 0.01 [c,d] | 0.15 ± 0.01 [c,d] | 29.36 ± 0.42 [a,b,c] | 29.38 ± 0.43 [a,b,c] |
| TVNu738 | 0.12 ± 0.01 [c,d] | 0.12 ± 0.01 [c,d] | 26.42 ± 0.39 [a,b,c,d] | 26.43 ± 0.40 [a,b,c,d] |
| TVNu1779 | 0.19 ± 0.02 [c,d] | 0.17 ± 0.01 [c,d] | 26.12 ± 2.04 [c,d,e] | 25.74 ± 2.01 [c,d,e] |
| TVNu605 | 0.42 ± 0.02 [c,d] | 0.36 ± 0.02 [c,d] | 29.16 ± 0.51 [a,b,c,d] | 29.18 ± 0.51 [a,b,c,d] |
| TVNu57 | 0.06 ± 0.01 [d] | 0.06 ± 0.01 [d] | 28.00 ± 0.55 [a,b,c,d] | 27.61 ± 0.48 [a,b,c,d] |
| TVNu138 | 0.23 ± 0.01 [c,d] | 0.21 ± 0.01 [c,d] | 27.19 ± 0.62 [a,b,c,d] | 26.80 ± 0.60 [a,b,c,d] |
| TVNu161 | 0.18 ± 0.01 [c,d] | 0.16 ± 0.01 [c,d] | 30.02 ± 0.77 [a,b] | 29.64 ± 0.76 [a,b] |
| TVNu758 | 0.16 ± 0.01 [c,d] | 0.15 ± 0.01 [c,d] | 27.10 ± 0.30 [a,b,c,d] | 27.11 ± 0.30 [a,b,c,d] |
| TVNu1825 | 0.25 ± 0.02 [c,d] | 0.23 ± 0.02 [c,d] | 25.50 ± 0.91 [d,e] | 25.51 ± 0.91 [d,e] |
| TVNu1522 | 0.19 ± 0.01 [c,d] | 0.17 ± 0.01 [c,d] | 22.56 ± 0.57 [e] | 22.57 ± 0.57 [e] |
| TVNu1388 | 0.18 ± 0.01 [c,d] | 0.16 ± 0.01 [c,d] | 26.53 ± 0.69 [a,b,c,d] | 26.54 ± 0.70 [a,b,c,d] |

**Table 5.** *Cont.*

| Species/Accession Number | Water Absorption Capacity | | Cooking Time (min) | |
|---|---|---|---|---|
| | Site A | Site B | Site A | Site B |

| Analysis of Variance (ANOVA) | | | | | | | | | | |
|---|---|---|---|---|---|---|---|---|---|---|
| | **Water Absorption Capacity** | | | | | **Cooking Time (min)** | | | | |
| **Source** | **DF** | **Sum of Squares** | **Mean Squares** | **F** | **p** | **DF** | **Sum of Squares** | **Mean Squares** | **F** | **p** |
| Model | 69 | 131.740 | 1.909 | 11.989 | <0.0001 | 69 | 22845.864 | 331.099 | 225.891 | <0.0001 |
| Error | 386 | 61.473 | 0.159 | | | 386 | 565.779 | 1.466 | | |
| Corrected Total | 455 | 193.213 | | | | 455 | 23411.643 | | | |

| Type III Sum of Squares Analysis | | | | | | | | | | |
|---|---|---|---|---|---|---|---|---|---|---|
| **Source** | **DF** | **Sum of Squares** | **Mean Squares** | **F** | **p** | **DF** | **Sum of Squares** | **Mean Squares** | **F** | **p** |
| Location (Site) | 2 | 0.025 | 0.013 | 0.080 | 0.924 | 2 | 0.052 | 0.026 | 0.018 | 0.982 |
| Genotype (Accessions) | 34 | 88.722 | 2.609 | 16.385 | <0.0001 | 34 | 12987.598 | 381.988 | 260.610 | <0.0001 |
| Location × Genotype | 33 | 0.033 | 0.001 | 0.006 | 1.000 | 33 | 0.000 | 0.000 | 0.000 | 1.000 |

Results are represented as the mean value of triplicates ± standard error. Mean values without any letter in common within each column are significantly different ($p = 0.05$). Site A: TARI-Selian; Site B: TaCRI. *DF*: Degree of freedom; *F*: *F*-ratio; *p*: *p*-value.

### 3.5. Cooking Time and Water Absorption of Vigna racemosa Accessions

The results for water absorption capacity and cooking time for accessions of wild *Vigna racemosa* are shown in Table 6. The values for water absorption capacity and cooking time tested showed no significant difference ($p > 0.05$) when compared with the values of their corresponding accession harvested in the other agro-ecological zone through two-way analysis of variance (ANOVA).

The water absorption capacity of some of the wild accessions showed significant difference to each other and to the three checks. The unknown *Vigna racemosa* and unknown *Vigna* legume accessions displayed significantly low values similar to the three checks (Table 6). The water absorption capacity of the wild *V. racemosa* accessions varied from 0.08 ± 0.01 to 1.35 ± 0.03 in site A and from 0.08 ± 0.00 to 1.32 ± 0.13 in site B (Table 6).

On the other hand, non-significant difference in cooking time between AGG51603WVIG1, AGG52867WVIG1 accessions and check 1 was observed. Besides, they were all significantly different from check 2, check 3, and the other accessions (Table 6). Generally, AGG53597WVIG1 exhibited superior low cooking time compared with the three checks. The cooking time for all accessions varied from 8.26 ± 0.42 to 30.33 ± 0.48 min in site A and from 7.87 ± 0.40 to 30.34 ± 0.50 min in site B (Table 6).

Pearson correlation analysis shows that there is a strong negative correlation between the water absorption and cooking time considering the wild *V. racemosa* accessions tested ($r = -0.91$ for site A and $r = -0.92$ for site B). Furthermore, the regression analysis shows that the water absorption capacity and cooking time are related by the equation: $Y = -17.17x + 32.10$ with $R^2 = 0.84$ (Figure 6)

### 3.6. Water Absorption Capacity, Cooking Time, and Clustering Analysis of the Four Vigna species for Domestication and Crop Improvement

The Figure 7 below shows the pattern of evolution of water absorption as a function of cooking time to depict the existing relationship between the two parameters for the eighty four accessions from the four wild *Vigna* species (*V. ambacensis*, *V. reticulata*, *V. vexillata*, and *V. racemosa*) and three domesticated species. It shows that the relationship is a strong negative correlation ($-0.69$ for site A and $-0.70$ for site B) between the water absorption and the cooking time which follows the equation: $Y_A = -7.99X + 26.52$ ($R^2 = 0.48$) or $Y_B = -8.21X + 26.57$ ($R^2 = 0.50$) (Figure 7).

Agglomerative Hierarchical Clustering (AHC) analysis performed on all the four *Vigna* species taking water absorption capacity, cooking time, and their individual weights before any processing as variable traits revealed seven classes (Figure 8). Details of various accessions belonging to each class are provided in Table 7. Class 1 consists of nineteen accessions of *V. reticulata*, sixteen accessions of *V. vexillata*, and all the eleven accessions of *V. ambacensis*. Class 2 consists of only eight accessions of *V. reticulata* and ten accessions of *V. vexillata* while class 3 consists of two accessions of *V. reticulata*, one accession of *V. vexillata*, three accessions of *V. racemosa* and check 2 and 3. The class 4 consists of one accession of *V. vexillata* and check 3 only, while one accession makes up class 5. Class 6 is made up of four accessions of *V. vexillata* and class 7 of two *V. reticulata* and two *V. vexillata*.

**Table 6.** Cooking time and water absorption capacity of *Vigna racemosa* accessions.

| Species/Accession Number | Water Absorption | | Cooking Time (min) | |
|---|---|---|---|---|
| | Site A | Site B | Site A | Site B |
| Landrace of *Vigna vexillata* | 1.33 ± 0.11 [a] | 1.32 ± 0.13 [a] | 10.24 ± 0.15 [d] | 10.26 ± 0.15 [d] |
| Cowpea (*Vigna unguiculata*) | 1.27 ± 0.08 [a] | 1.27 ± 0.08 [a] | 16.29 ± 0.15 [b] | 16.31 ± 0.15 [b] |
| Rice Bean (*Vigna umbellata*) | 1.16 ± 0.06 [a] | 1.16 ± 0.06 [a] | 13.20 ± 0.12 [c] | 13.23 ± 0.12 [c] |
| AGG53597WVIG1 | 1.35 ± 0.03 [a] | 1.33 ± 0.02 [a] | 8.26 ± 0.42 [d] | 7.87 ± 0.40 [d] |
| AGG51603WVIG1 | 1.29 ± 0.01 [a] | 1.27 ± 0.02 [a] | 10.15 ± 0.22 [d,e] | 10.17 ± 0.25 [d,e] |
| AGG52867WVIG1 | 1.04 ± 0.04 [a] | 1.02 ± 0.00 [a] | 11.27 ± 0.41 [d] | 11.28 ± 0.42 [d] |
| Unknown *Vigna* legume | 0.43 ± 0.01 [a,b] | 0.39 ± 0.02 [a,b] | 29.35 ± 0.31 [a] | 28.97 ± 0.30 [a] |
| Unknown *Vigna racemosa* | 0.08 ± 0.01 [b] | 0.08 ± 0.00 [b] | 30.33 ± 0.48 [a] | 30.34 ± 0.50 [a] |

**Analysis of Variance (ANOVA)**

| | Water Absorption Capacity | | | | | Cooking Time (min) | | | | |
|---|---|---|---|---|---|---|---|---|---|---|
| Source | *DF* | Sum of Squares | Mean Squares | *F* | *p* | *DF* | Sum of Squares | Mean Squares | *F* | *p* |
| Model | 15 | 13.441 | 0.896 | 4.279 | <0.0001 | 15 | 4957.993 | 330.533 | 386.632 | <0.0001 |
| Error | 278 | 58.223 | 0.209 | | | 278 | 237.663 | 0.855 | | |
| Corrected Total | 293 | 71.664 | | | | 293 | 5195.656 | | | |

**Type III Sum of Squares Analysis**

| Source | *DF* | Sum of Squares | Mean Squares | *F* | *p* | *DF* | Sum of Squares | Mean Squares | *F* | *p* |
|---|---|---|---|---|---|---|---|---|---|---|
| Location (Site) | 1 | 0.004 | 0.004 | 0.017 | 0.896 | 1 | 0.006 | 0.006 | 0.007 | 0.934 |
| Genotypes (Accessions) | 7 | 13.436 | 1.919 | 9.165 | <0.0001 | 7 | 4957.947 | 708.278 | 828.489 | <0.0001 |
| Location × Genotype | 7 | 0.003 | 0.000 | 0.002 | 1.000 | 7 | 0.001 | 0.000 | 0.000 | 1.000 |

Results are represented as the mean value of triplicates ± standard error. Mean values without any letter in common within each column are significantly different ($p = 0.05$). Site A: TARI-Selian; Site B: TaCRI. *DF*: Degree of freedom; *F*: *F*-ratio; *p*: *p*-value.

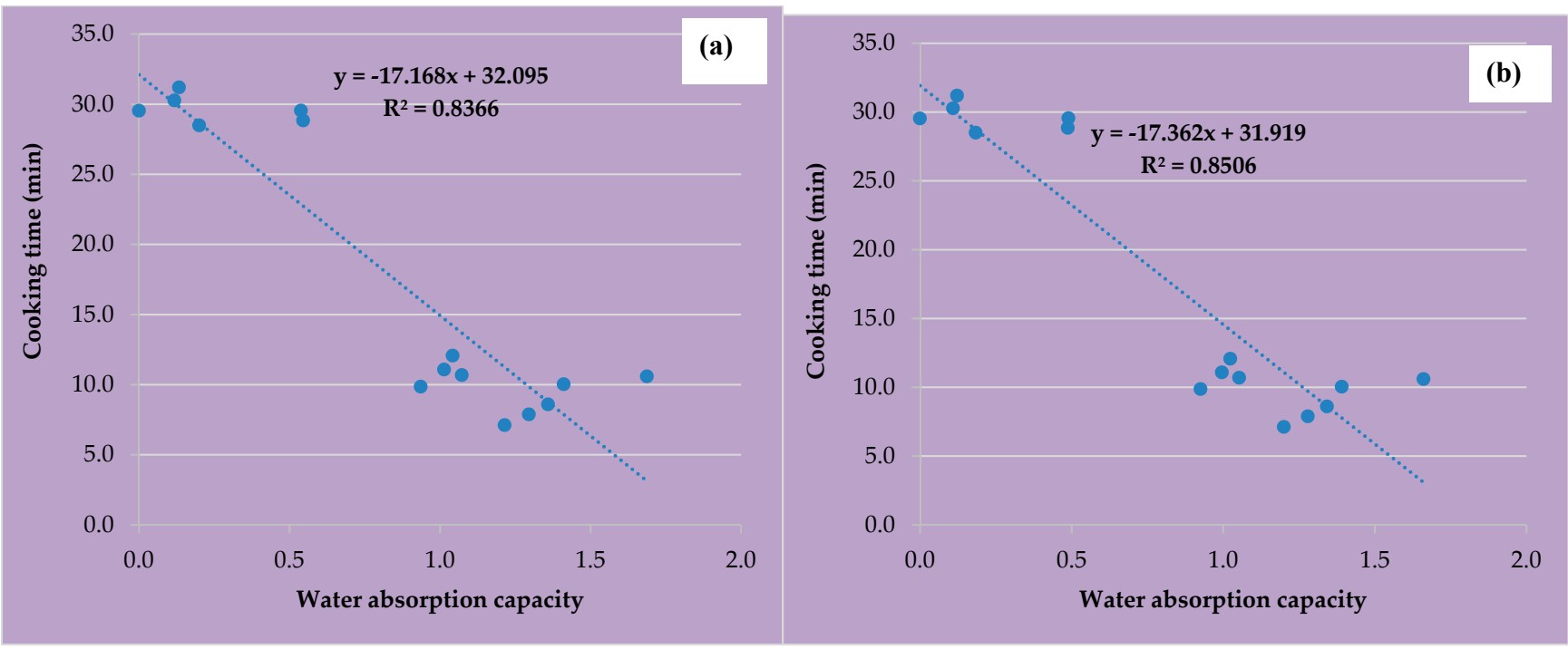

**Figure 6.** Correlation between water absorption and cooking time for the *Vigna racemosa* accessions. (**a**) Plotted with data from Site A; (**b**) plotted with data from Site B.

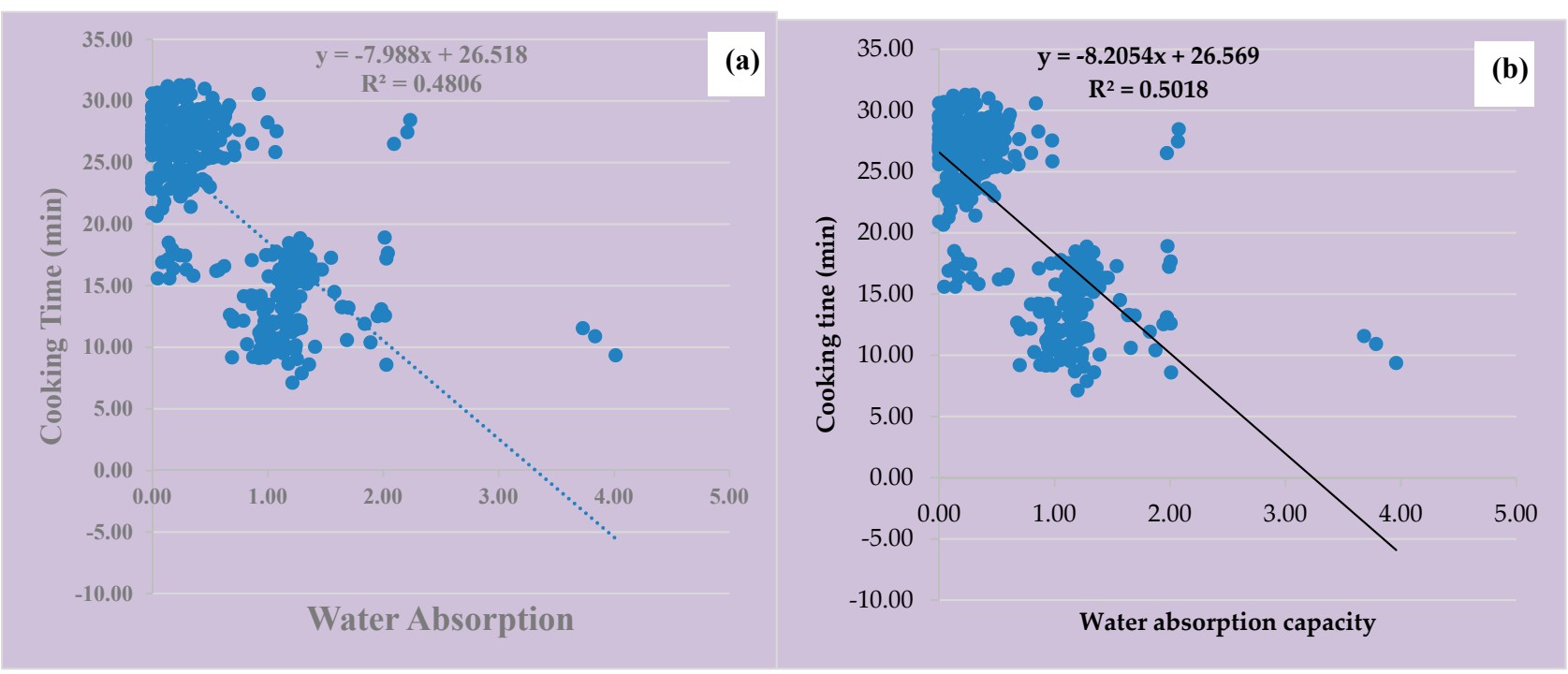

**Figure 7.** Correlation between water absorption and cooking time for the *Vigna* species studied. (**a**) Plotted with data from Site A; (**b**) plotted with data from Site B.

**Table 7.** Details of classes from the dendrogram *.

| Class | 1 | 2 | 3 | 4 | 5 | 6 | 7 |
|---|---|---|---|---|---|---|---|
| **Object** | 47 | 20 | 8 | 2 | 1 | 4 | 4 |
| | TVNu324_VRe | TVNu1632_VV | TVNu325_VRe | Check 3 | TVNu1520_VRe | AGG308107WVIG2_VV | TVNu379_VRe |
| | TVNu342_VA | TVNu1701_VV | Check 2 | TVNu781_VV | | AGG62154WVIG_1_VV | TVNu1582_VV |
| | AGG308101WVIG1 _VV | TVNu1629_VV | Unknown _*Vigna reticulata* | | | TVNu1624_VV | TVNu1358_VV |
| | TVNu1344_VV | TVNu767_VRe | AGG51603WVIG1_VRa | | | AGG308097WVIG1_VV | TVNu1394_VRe |
| | AGG308096 WVIG2_VV | TVNu343_VRe | AGG53597WVIG1_VRa | | | | |
| | TVNu120 _VV | TVNu333_VV | Check 1 | | | | |
| | TVNu1529_VV | TVNu1370_VV | TVNu837_VV | | | | |
| | TVNu720_VA | TVNu349_VRe | AGG52867WVIG1_VRa | | | | |
| | TVNu223_VA | TVNu1378_VV | | | | | |
| | TVNu1546_VV | TVNu1405_VRe | | | | | |
| | TVNu1698_VRe | TVNu1593_VV | | | | | |
| | TVNu877_VA | Unknown *Vigna* | | | | | |
| | TVNu524_VRe | TVNu381_VV | | | | | |
| | TVNu1699_VA | TVNu479_VV | | | | | |
| | TVNu1191_VRe | TVNu605 _VRe | | | | | |
| | TVNu1621_VV | TVNu1360_VV | | | | | |
| | TVNu607_VRe | TVNu1790_VRe | | | | | |
| | TVNu56_VRe | TVNu1808_VRe | | | | | |
| | TVNu- 224_VRe | Unknown_*Vigna_racemosa* | | | | | |
| | TVNu739_VRe | TVNu161_VRe | | | | | |
| | TVNu916_VRe | | | | | | |
| | TVNu955_VV | | | | | | |
| | TVNu1092 _VV | | | | | | |
| | TVNu1591_VV | | | | | | |
| | TVNu178_VV | | | | | | |
| | TVNu293_VV | | | | | | |
| | TVNu1840_VA | | | | | | |

**Table 7.** *Cont.*

| Class | 1 | 2 | 3 | 4 | 5 | 6 | 7 |
|---|---|---|---|---|---|---|---|
| **Object** | 47 | 20 | 8 | 2 | 1 | 4 | 4 |
| | AGG17856WVIG_1_VRe | | | | | | |
| | TVNu738_VRe | | | | | | |
| | TVNu1796_VV | | | | | | |
| | TVNu1792_VA | | | | | | |
| | TVNu832_VV | | | | | | |
| | TVNu219_VA | | | | | | |
| | TVNu491_VRe | | | | | | |
| | TVNu1628_VV | | | | | | |
| | TVNu1779_VRe | | | | | | |
| | TVNu138_VRe | | | | | | |
| | AGG308099WVIG2_VV | | | | | | |
| | TVNu1804_VA | | | | | | |
| | TVNu1586_VV | | | | | | |
| | TVNu57_VRe | | | | | | |
| | TVNu1825_VRe | | | | | | |
| | TVNu1644_VA | | | | | | |
| | TVNu758_VRe | | | | | | |
| | TVNu1388_VRe | | | | | | |
| | TVNu1522_VRe | | | | | | |
| | TVNu1185_VA | | | | | | |

* Abbreviations put beside the accession names serves to identify species: VA stands for *V. ambacensis*, VV for *V.vexillata*, VRe for *V. reticulata*, and VRa for *V. racemos*.

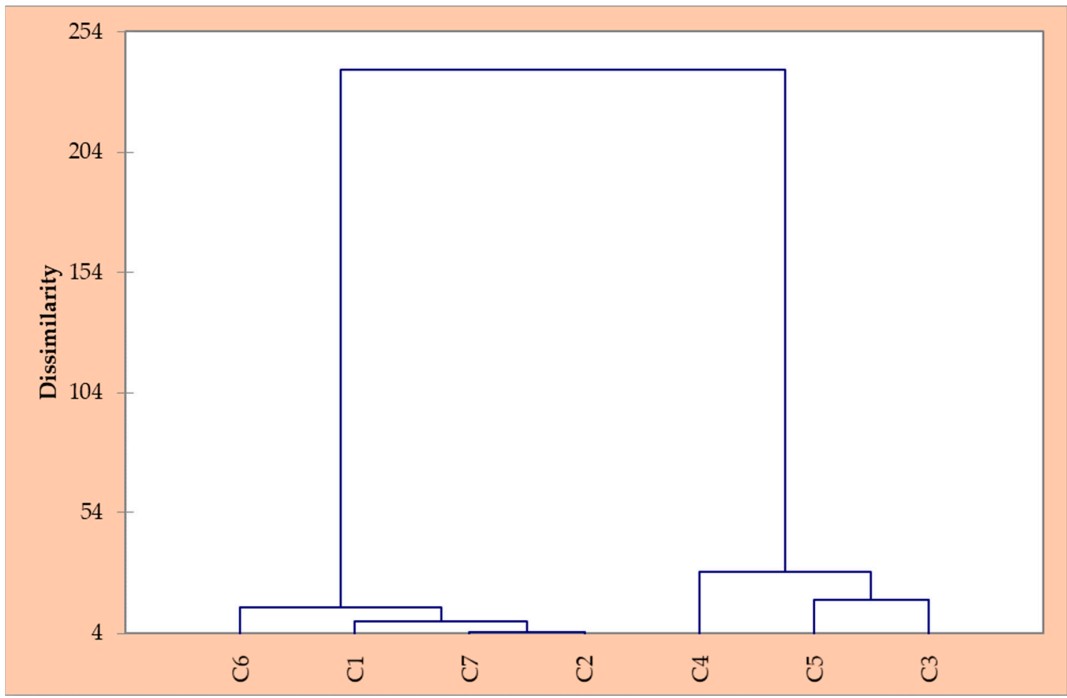

**Figure 8.** Dendrograms showing relationship among 84 accessions of the four wild *Vigna* species and three domesticated varieties regarding their weights before soaking, water absorption and cooking time.

*3.7. Descriptive Statistics and Yield Traits of the Wild Vigna Species*

Table 8 shows results of the means values for water absorption capacity, cooking time, and yield traits of the four wild species studied. *Vigna ambacensis* present mean values of 0.20, 23.45 min, and 1.74 g for water absorption capacity, cooking time, and yield per plant, respectively, in site A, while in site B the mean values are 0.18, 23.43 min, and 0.78 g for water absorption capacity, cooking time, and yield per plant, respectively. In *Vigna vexillata*, the values of 0.34, 25.42 min, and 16.84 g were found for water absorption capacity, cooking time, and yield per plant, respectively, in site A and 0.32, 25.40 min, and 12.54 g for water absorption capacity, cooking time, and yield per plant, respectively, in site B. For *Vigna reticulata,* the mean values are 0.39, 26.77 min, and 10.60 g for water absorption capacity, cooking time, and yield per plant, respectively, in site A, while in site B the mean values are 0.37, 26.78 min, and 6.78 g for water absorption capacity, cooking time, and yield per plant, respectively. Finally, *Vigna racemosa* present mean values of 0.84, 17.70 min, and 28.25g for water absorption capacity, cooking time, and yield per plant, respectively, in site A, while in site B the mean values are 0.81, 17.72 min, and 18.28 g for water absorption capacity, cooking time, and yield per plant, respectively. The yield values varied from 13.45 g (*V. vexillata* landrace) to 86.04 g (rice bean) in site A, while it varied from 7.62 g (*V. vexillata* landrace) to 61.92 g (rice bean) in site B for the domesticated legumes. For the wild legumes, it varied from 1.74 g (*Vigna ambacensis*) to 28.25 g (*Vigna racemosa*) in site A and from 0.78 g (*Vigna ambacensis*) to 18.28 g (*Vigna racemosa*) in site B.

**Table 8.** Descriptive statistic and yield traits of the wild *Vigna* species.

| Species | Descriptive Parameters | Water Absorption Capacity | | Cooking Time (min) | | Yield per Plant (g) | |
|---|---|---|---|---|---|---|---|
| | | Site A | Site B | Site A | Site B | Site A | Site B |
| Landrace of *Vigna vexillata* | Mean | 1.33 | 1.32 | 10.24 | 10.26 | 13.45 | 7.62 |
| | CV (%) | 9.50 | 9.37 | 1.46 | 1.45 | 4.59 | 6.34 |
| | Range | 0.69–4.01 | 0.70–3.96 | 8.56–11.89 | 8.59–11.91 | 9.00–26.55 | 4.94–17.31 |
| Cowpea (*Vigna unguiculata*) | Mean | 1.27 | 1.27 | 16.29 | 16.31 | 52.690 | 26.657 |
| | CV (%) | 1.85 | 1.82 | 0.93 | 0.93 | 5.48 | 5.42 |
| | Range | 0.58–1.58 | 0.58–1.57 | 14.06–18.84 | 14.09–18.87 | 28.80–106.08 | 14.71–53.35 |
| Rice Bean (*Vigna umbellata*) | Mean | 1.16 | 1.16 | 13.20 | 13.23 | 86.04 | 61.92 |
| | CV (%) | 4.05 | 4.02 | 0.92 | 0.91 | 2.378 | 2.361 |
| | Range | 0.67–2.02 | 0.68–2.00 | 11.73–14.98 | 11.76–15.01 | 60.27–109.76 | 43.51–78.86 |
| *Vigna ambacensis* | Mean | 0.20 | 0.18 | 23.45 | 23.43 | 1.74 | 0.78 |
| | CV (%) | 11.21 | 11.20 | 0.44 | 0.42 | 22.36 | 14.25 |
| | Range | 0.00–0.50 | 0.00–0.58 | 22.25–24.95 | 22.26–24.96 | 0.72–5.36 | 0.43–1.65 |
| *Vigna vexillata* | Mean | 0.34 | 0.32 | 25.42 | 25.40 | 16.84 | 12.54 |
| | CV (%) | 7.80 | 7.95 | 1.73 | 1.70 | 9.48 | 6.77 |
| | Range | 0.00–1.15 | 0.00–1.13 | 15.54–31.28 | 15.55–31.30 | 9.48–63.00 | 7.61–35.26 |
| *Vigna reticulata* | Mean | 0.39 | 0.37 | 26.77 | 26.78 | 10.60 | 6.78 |
| | CV (%) | 13.83 | 14.04 | 1.20 | 1.27 | 10.55 | 10.62 |
| | Range | 0.00–2.24 | 0.00–2.08 | 16.60–30.98 | 16.58–40.00 | 4.32–30.36 | 2.58–17.69 |
| *Vigna racemosa* | Mean | 0.84 | 0.81 | 17.70 | 17.72 | 28.25 | 18.28 |
| | CV (%) | 16.70 | 17.06 | 14.83 | 14.81 | 37.02 | 38.37 |
| | Range | 0.00–1.69 | 0.00–1.66 | 7.11–31.19 | 7.12–31.22 | 2.08–49.00 | 1.21–34.70 |

CV: Coefficient of variation; Range (Minimum−Maximum).

## 4. Discussion

The values for water absorption capacity and cooking time showed no significant difference when compared with the values of their corresponding accessions harvested in the other agroecological zone for all the accessions tested. This could be due to the existence of a very slight difference in the characteristics of the two agroecological zones that could not significantly affect the genetic performance of the *Vigna* genus regarding the weight, water absorption, and cooking time. This is further justified by the fact that the interaction effect (location × genotype) showed that the differences observed for cooking time and water absorption capacities do not depend on location in all the accessions tested in this study (Tables 2a, 3, 4, 5 and 6). In the same line, a recent report revealed that the agroecological conditions could affect some nutrients like amino acids, protein, and minerals in quinoa but have no effect on their saponin and fiber content [24]. Furthermore, this study also demonstrates that the replication of the same species within the same location does not depend on the other species for the water absorption capacity trait (Table 2b), while for cooking time trait, there is an interaction with other species within the same location (Table 2c). This could be an important characteristic to be exploited in breeding programs.

The non-significant or significant changes observed in the mean seed weights of some accessions when compared before and after soaking depicted here by their water absorption capacity values could be explained by the fact that some accessions possess a seed coat more water permeable than others (Tables 2–6). The seed coat water permeability of seeds as a phenotype possesses a crucial role in legumes cooking properties and germination [25]. However, the development of legume seed coat has not yet been characterized at a molecular level to strongly support its genetic implication [26]. A study involving legume showed that the water absorption of dry beans differs between varieties [27].

Looking at the *V. ambacensis* species, all the wild accessions exhibited significantly lower water absorption capacity values as compared with all three checks (Table 3). The accession TVNu342, with a water absorption capacity not significantly different from the checks exhibited a higher cooking time. This could imply that not only the water absorption capacity is directly or indirectly linked to cooking times of legumes and requires further physiological investigation. The genus *Vigna* possess a very large number of species in which very few have been studied extensively. The *V. ambacensis* is among the non-studied species [2]. The very first comprehensive web genomic resource of the genus *Vigna* has just recently been published and that covered only three commercially domesticated species [28]. Taxonomic rearrangements are also still under investigation [29] and efforts to domesticate some of the selected wild *Vigna* species is in progress [2,4]. Pearson correlation analysis shows that there is no correlation between the water absorption and cooking time considering only the three domesticated species ($r = -0.025$). This could be due to some individual physiological differences or similarities among the tested accessions which requires further examination at molecular level as reports on *V. ambacensis* studies are very scanty and need to be addressed for proper exploitation of its full potential towards domestication [2].

For the *V. vexillata* species, Table 4 proved that there are some phenotypic similarities between the wild accessions with the cowpea and rice bean with regard to their water absorption capacity values as many accessions show no significant different values with those two checks. Henceforth, it requires further investigations at molecular level involving phylogenetic analysis to establish a strong relationship between the accessions. In this regard, it is noted that the genetic diversity and structure of *V. vexillata* as well as many wild *Vigna* legumes are still under investigation [2,29–31]. The idea is also supported by an earlier report that stipulated that domestication of the commercial *V. vexillata* (zombie pea) is not certain and it took place more than once in different regions [32]. Concerning the cooking time (Table 4), there is a high diversity in differences among the accessions. This could also explain why there is a weak negative correlation between the water absorption and cooking time considering the wild *V. vexillata* tested ($r = -0.31$) (Figure 4).

Wild *V. reticulata* species revealed that there is no significant difference between accessions regarding water absorption capacity (Table 5) with cowpea and rice bean except for TVNu1520, TVNu325 and the *V. vexillata* landrace. This demonstrates a considerable variability among the accessions as far as water absorption capacity is concerned as a phenotypic trait. Considering the cooking time (Table 5), a high diversity in differences of means among the accessions is noticed. Twenty-five accessions show no significant difference to each other but significantly different from the *V. vexillata* landrace and rice bean, while cowpea showed no significant difference to TVNu325 and the unknown *V. reticulata* accession. Curiously, scanty information about *V. reticulata* is also noticed. The genotype interactions in both water absorption capacity and cooking time phenotypic traits simply demonstrate the phenotypic diversity of these accessions which is very important in breeding.

Though very few accessions were included in this study, *V. racemosa* species present more phenotypic similarities with the *V. vexillata* landrace and cowpea with respect to the water absorption capacity and cooking time traits studied. It was revealed from the results that there is no significant difference between the means of the following wild accessions (AGG51603WVIG1, AGG53597WVIG1, AGG52867WVIG1) regarding their weights before soaking and the *V. vexillata* landrace and cowpea. The weights taken after the soaking process revealed a similar phenomenon while the water absorption shows closeness to rice bean. In the case of cooking time, two accessions seem to be related to the *V. vexillata* landrace. All these assumptions need further investigations as *V. racemosa* also suffer from scanty information.

This study also showed that there is a strong negative correlation between the water absorption and the cooking time with a correlation coefficient of r = $-0.69$ which follows the equation: $Y = -7.99x + 26.52$ ($R^2 = 0.48$) for site A and $Y = -8.21x + 26.57$ ($R^2 = 0.50$) for the site B (Figure 7). This result is in line with previous reports. For example, an early report proved that the cooking time was longer in bean varieties without prior soaking [33]. A similar result was found within classes of oriental

noodle, in which cooking time was significantly shortened with increase in water absorption [34]. This could be an important parameter to guide the breeding of legumes with regards to cooking time when knowing their water absorption capacity.

Agglomerative hierarchical clustering (AHC) analysis performed on all the four species revealed the existence of seven classes when weight of accessions before the soaking process, water absorption capacity, and cooking time are taken as parameters (Figure 8). Details of various accessions belonging to each class are provided in Table 7. The analysis shows that some accessions between the four species can be grouped together in the same cluster as they present similar traits or relationship. It is in line with what the first comparison based on Tukey analysis showed in this study. For example, class 1 consists of *V. vexillata*, *ambacensis* and *reticulata* accessions while class 2 is mainly *V. vexillata* with few *V. reticulata*. It is also noted that all *V. ambacensis* are grouped in class 1. This can simply imply that there are phenotypic trait similarities of the accessions within species with each other and with checks. However, further molecular investigations are needed to fully investigate assumptions of any genetic relationship within and between species. The classifications of the *Vigna* species remain a continuous and evolving process as their origin are still subject of speculations. For example, it is reported that the Asian *Vigna* were still belonging to the genus *Phaseolus* until 1970 [30]. It is generally speculated that the *Vigna* might have originated from Africa and evolved from the African genus *Wajira* as it is basal compared with *Vigna* and *Phaseolus* [30]. Although, little attention has been paid to the conservation of the African wild *Vigna* species as more than 20 species are apparently not conserved in any ex-situ collection despite their several ethnobotanical uses [30]. Therefore, it could be speculated from this study that accessions in groups 3, 4, and 5 are likely candidates for domestication since these groups contain the check lines, though further investigations are required.

Based on a general assessment view, the values of yield per plant for the wild *Vigna* species studied here are lower than those of the domesticated species, especially cowpea and rice bean (Table 8). A similar finding was reported by an earlier report [7]. However, it might be important to note that the yield per plant for these wild legume accessions may be influenced by their seed characteristics because some of them could have a high number of seeds per plant with a surprising low weight as compared with the domesticated ones that produced fewer numbers of seed. The low seed weights in wild accessions could be attributed to their small seed sizes compared to domesticated ones with bigger seed sizes. The domesticated species here could have certainly acquired bigger seed sizes during the domestication process. Seed size is one of the important domestication traits [35] that should be considered by breeders in the course of improvement and domestication of these wild legumes as they all presented smaller seed sizes by mere looking (Figure 1). From this study, yield, water absorption capacity, and cooking time are apparently not related, though they are very important traits that need to be considered in breeding and selection of wild candidates for domestication. This may be due to the fact that yield mainly depends on seed physical characteristics such as seed size, seed weight, and seed number, while cooking time and water absorption capacity depends on seed physiological characteristics such as seed coat biosynthesis [25]. This could also be supported by the high variation in yield between locations as compared with low variations in cooking time and water absorption capacity (Table 8). In the same vein, it is also noted that the domesticated legumes possess high values of yield per plant in addition to their low cooking time and high water absorption capacity values as compared with the wild ones. Such characteristics might be among the factors that hinders their utilization as earlier reported [4]. Yield is a very important trait in crop domestication. However, these wild legumes with multipurpose utilizations as suggested by farmers in our earlier investigation [4] fit well as candidates for domestication considering the domestication criteria established by researchers recently [35]. Crop domestication of novel species is becoming one of the potential alternatives to mitigate the global food security challenge.

## 5. Conclusions

Despite their under-exploitation for human benefits, the wild *Vigna* legumes possess important cooking characteristics comparable with the domesticated ones. The present study revealed that the cooking time and water absorption capacity of wild legumes do not depend on their cultivation environment. Furthermore, it proved that there is a strong negative correlation between the water absorption capacity and cooking time in wild *Vigna* species. The study also revealed that some wild *Vigna* species present no significant difference in their cooking times with domesticated species which could be a positive acceptability trait to consumers. However, they might require considerable improvement in terms of seed physical characteristics to impact on their yield. Such key preliminary information could be of vital consideration in breeding, improvement, and domestication of wild *Vigna* legumes to make them useful for human benefit as far as cooking time is concerned. Investigations of nutritional and biochemical composition of these under-exploited legumes will also be of great importance to both scientists (breeders) and consumers for achieving food variety addition.

**Author Contributions:** P.A.N. and A.O.M conceived and designed the experiments; D.V.H. performed the experiments, collected data, analyzed the data, and made the first draft of the manuscript; P.B.V. and A.O.M. supervised the research and internally reviewed the manuscript; and P.A.N. made the final internal review and revised the final draft manuscript.

**Funding:** This research was partially funded by the Centre for Research, Agricultural Advancement, Teaching Excellence and Sustainability in Food and Nutrition Security (CREATES-FNS) through the Nelson Mandela African Institution of Science and Technology (NM-AIST). The research also received funding support from the International Foundation for Science (IFS) through the grant number I-3-B-6203-1.

**Acknowledgments:** This research was partially funded by the Centre for Research, Agricultural Advancement, Teaching Excellence and Sustainability in Food and Nutrition Security (CREATES-FNS) through the Nelson Mandela African Institution of Science and Technology (NM-AIST). The authors acknowledge the additional funding support from the International Foundation for Science (IFS) through the grant number I-3-B-6203-1. The authors also acknowledge the Genetic Resources Center, International Institute of Tropical Agriculture (IITA), Ibadan-Nigeria as well as the Australian Grains Genebank (AGG) for providing detail information and seed materials for illustration. The authors are thankful to Mary Mdachi and Lameck Makoye of the Tanzania Agricultural Research Institute (TARI), Selian-Arusha, Tanzania for their technical support in carrying out the Mattson Cooking experiment. The statistical advice from Usman Rachid of the Health and Rehabilitation Research Center, Auckland University of Technology, New Zealand is also appreciated.

**Conflicts of Interest:** The authors declare no conflict of interest.

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
