# Peer review of "Assessment of Water Absorption Capacity and Cooking Time of Wild Under-Exploited Vigna Species towards their Domestication"

_agronomy, doi:10.3390/agronomy9090509_

Round 1
Reviewer 1 Report
Vigna is an important legume as a source of staple protein in several parts of Africa. Characterization of neglected variability / germplasm within the genus is welcome, as it may help current and future breeding programs. However, "cooking time" is a minor phenotype as compared to yield, composition, digestibility, or organoleptic properties and these should have been characterized first. While experimentation is generally ably performed, I do not believe authors make a convincing case of why a study such as this is relevant, timely, or sufficiently advances the state of the art.
Minor queries are:
1. The first sentence in Abstract is incomprehensible
2. Statement in l. 41 is unsupported by data
3. Fig 1 shows photographs and not microphotographs
4. Several errors appear in the references section
5. l. 304, section 3.7 I see no genetic implications
6. Statements in l. 346, l. 361-363, l. 364-370, l. 378-379 are unsupported.
7. l. 348-354 are incomprehensible
Author Response
REVIEWER #1 COMMENTS AND RESPONSES:
Comments and Suggestions for Authors
Vigna is an important legume as a source of staple protein in several parts of Africa. Characterization of neglected variability / germplasm within the genus is welcome, as it may help current and future breeding programs. However, "cooking time" is a minor phenotype as compared to yield, composition, digestibility, or organoleptic properties and these should have been characterized first. While experimentation is generally ably performed, I do not believe authors make a convincing case of why a study such as this is relevant, timely, or sufficiently advances the state of the art.
Response:
The authors agree with the reviewer that the Vigna genus is an important legume as a source of staple protein in several parts of Africa and that the characterization of neglected variability / germplasm within the genus is welcome, as it may help current and future breeding programs. Therefore, in addition to the cooking time phenotype data, authors have added yield traits characteristics of the wild Vigna on page 37 and discussed on page 40, lines 532- 534 as recommended.
Minor queries are:
Point 1: The first sentence in Abstract is incomprehensible
Response 1: The statement has been rephrased as recommended, refer to page 1 (line 17-18).
Point 2: Statement in l. 41 is unsupported by data
Response 2: The unsupported statements have been deleted (refer to page 6, line 168- 169 &172- 173).
Point 3: Fig 1 shows photographs and not microphotographs
Response 3: The word micrograph has been corrected as photograph accordingly (refer to page 5, line 150).
Point 4: Several errors appear in the references section
Response 4: Errors in reference section have been corrected (see reference 1, 2, 4, 5, 10, 12, 17, 26, 27 Pages 40- 43).
Point 5: l. 304, section 3.7 I see no genetic implications
Response 5: The concerned section title (line 376-377, section 3.7) has been rephrased to suit the content as recommended (refer to page 31, lines 376- 377).
Point 6: Statements in l. 346, l. 361-363, l. 364-370, l. 378-379 are unsupported.
Response 6:
Statements in:
- l. 346, the statement has been improved and supported (refer to page 38, lines 445- 458)
- l. 361-363, the statement has been improved and supported (refer to page 38, lines 465- 468)
- l. 364-370, the statement has been improved and supported (refer to page 38, lines 470- 478)
- l. 378-379, the statement has been improved (refer to page 39, lines 486- 489)
Point 7: l. 348-354 are incomprehensible
Response 7: Statements on lines 348- 354 have been revised and improved (refer to page 38, lines 450- 458).

Reviewer 2 Report
This manuscript addressed an important investigation on determination of cooking time and the water absorption in 80 wild Vigna legumes in addition to their relationship. The authors applied various processes to analyze the data. The authors found variation of cooking time and water absorption among several genotypes. Twenty-five wild genotype of Vigna accessions showed no interaction between the cooking time and the water absorption capacity when tested. The authors further demonstrated that there is a strong negative correlation between the water absorption capacity and cooking time in some of the wild Vigna species which are cross compatible with cultivated varieties. This results will be useful for breeders to produce value added Vigna species.
The abstract part should highlight some of the key results with more clarity.
Page 3, line 100-102, Revise it; line 111, provide accessions numbers, line 114, replace sundried into sun-dried.
Results and discussion is confusing and has lot of repetitions. Revise it.
Author Response
REVIEWER #2 COMMENTS AND RESPONSES:
Comments and Suggestions for Authors
This manuscript addressed an important investigation on determination of cooking time and the water absorption in 80 wild Vigna legumes in addition to their relationship. The authors applied various processes to analyze the data. The authors found variation of cooking time and water absorption among several genotypes. Twenty-five wild genotype of Vigna accessions showed no interaction between the cooking time and the water absorption capacity when tested. The authors further demonstrated that there is a strong negative correlation between the water absorption capacity and cooking time in some of the wild Vigna species which are cross compatible with cultivated varieties. This results will be useful for breeders to produce value added Vigna species.
Point 1: The abstract part should highlight some of the key results with more clarity.
Response 1: The abstract section has been improved accordingly (refer to page 1, lines 17- 38).
Point 2: Page 3, line 100-102, Revise it; line 111, provide accessions numbers, line 114, replace sundried into sun-dried.
Response 2:
- Page 3, line 100-102; the statement has been revised (refer to page 3, lines 115-117)
- line 111, Accession number are provided in all results tables (see tables 3[page13], 4[page17], 5[page24], 6[page28]).
- line 114, the spelling of the word ‘’sundried’’ has been corrected accordingly (see page 3 line 129).
Point 3: Results and discussion is confusing and has lot of repetitions. Revise it.
Response 3: Results and discussion sections have been improved (refer to page 7- 39)

Reviewer 3 Report
I have a number of concerns with statistical analyses and presentation. In Table 2, you should also show the replicate within location effect in this part of the table. You could also report on triplicate within samples to see whether the effort of measuring triplicates would be useful in a plant breeding program for selection for these traits.
In Fig. 2 and 3, are these regressions significant? It looks unlikely. If the regression is not significant, you should not show the regression line.
In Table 4, how can the values for TVNU1378, 381 and 178 all have a significance letter of d when TVNU591 has significance of a,b,c,d. Something is wrong here. Also it seems the DF column is missing and all need to be shifted over to the left.
On line 258 and Fig. 6, if there is no significant site effect, you can discuss means over sites.
There are several instances of the conclusions overreaching the work reported in the manuscript. Line 364, you cannot comment on genetics since you only phenotyped the lines. Line 385, you cannot comment on relatedness only on similarity between the traits you measured. Line 408, you cannot comment on cross compatibility only similarity with the traits you measured.
Other comments have been made on the attached manuscript.

Author Response
REVIEWER #3 COMMENTS AND RESPONSES:
Comments and Suggestions for Authors
Point 1: I have a number of concerns with statistical analyses and presentation. In Table 2, you should also show the replicate within location effect in this part of the table. You could also report on triplicate within samples to see whether the effort of measuring triplicates would be useful in a plant breeding program for selection for these traits.
Response 1: The details on replicate within location interactions have included as recommended (refer to page 10-11 [Table 2b& 2c], Lines 224- 229 of page 7-8 for description and lines 437-441 of page 38 for discussion).
Point 2: In Fig. 2 and 3, are these regressions significant? It looks unlikely. If the regression is not significant, you should not show the regression line.
Response 2: The regression lines on Fig.2 and 3 have been removed accordingly (refer to page 12 and page 15).
Point 3: In Table 4, how can the values for TVNU1378, 381 and 178 all have a significance letter of d when TVNU591 has significance of a,b,c,d. Something is wrong here. Also it seems the DF column is missing and all need to be shifted over to the left.
Response 3: The authors thank the reviewer for this pertinent remark. The authors have checked the data and realized some mistakes performed during the editing of the table. The table has been reorganized and improved to eliminate the pointed statistical incoherence. The DF column has also been included and the part of the table shifted to the left as recommended (refer to pages 17- 20).
Point 4: On line 258 and Fig. 6, if there is no significant site effect, you can discuss means over sites.
Response 4: The means over site have been discussed accordingly and included in the revised manuscript at page 23, lines 329- 331 and page 38, lines 493- 495.
Point 5: There are several instances of the conclusions overreaching the work reported in the manuscript. Line 364, you cannot comment on genetics since you only phenotyped the lines. Line 385, you cannot comment on relatedness only on similarity between the traits you measured. Line 408, you cannot comment on cross compatibility only similarity with the traits you measured.
Response 5:
- Line 364, the statement has been improved accordingly (refer to page 38, lines 470-472).
- Line 385, the statement has been improved accordingly (refer to page 39, lines 496-498).
- Line 408, the statement and claims/assumptions have been improved accordingly (refer to page 39, lines 519-523).
Point 6: Other comments have been made on the attached manuscript.
Response 6: The other comments made on the attached manuscript have been addressed accordingly (refer to page 5: Fig. 1 where the ruler has been added to the figure; Page28 [line357], page 31[line 382], page 39: lines 529-492531).
